# POLYCHROMIC OBJECTIVES FOR REINFORCEMENT LEARNING

**Jubayer Ibn Hamid**\*, **Ifdita Hasan Orney**\*, **Ellen Xu, Chelsea Finn, Dorsa Sadigh**
Stanford University

## ABSTRACT

Reinforcement learning fine-tuning (RLFT) is a dominant paradigm for improving pretrained policies for downstream tasks. These pretrained policies, trained on large datasets, produce generations with a broad range of promising but unrefined behaviors. Often, a critical failure mode of RLFT arises when policies lose this diversity and collapse into a handful of easily exploitable outputs. This convergence hinders exploration, which is essential for expanding the capabilities of the pretrained policy and for amplifying the benefits of test-time compute scaling. To address this, we introduce an objective for policy gradient methods that explicitly enforces the exploration and refinement of diverse generations, which we call a polychromic objective. We then show how proximal policy optimization (PPO) can be adapted to optimize this objective. Our method (1) employs vine sampling to collect on-policy rollouts and (2) modifies the advantage function to reflect the advantage under our new objective. Experiments on BabyAI, Minigrid, and Algorithmic Creativity show that our method improves success rates by reliably solving a larger set of environment configurations and generalizes better under large perturbations. Moreover, when given multiple attempts in pass@$k$ experiments, the policy achieves substantially higher coverage, demonstrating its ability to maintain and exploit a diverse repertoire of strategies.

## 1 INTRODUCTION

Reinforcement learning fine-tuning (RLFT) is widely used to enhance the performance of pretrained models across diverse downstream domains. For instance, RLFT has been applied to steer large language models (LLMs) toward instruction following and complex reasoning (DeepSeek-AI et al., 2025; OpenAI et al., 2024; Qwen, 2025). A common thread across these settings is the availability of expressive generative models (i.e., pretrained distributions), trained on large and diverse datasets, that already exhibit a broad repertoire of strategies. RLFT then refines these distributions by reinforcing the strategies that yield higher reliability and performance.

However, exploration during RLFT remains a central challenge. Prior work (Cui et al., 2025; Zhao et al., 2025) has documented entropy collapse: instead of expanding their repertoire, fine-tuned policies concentrate probability mass on a narrow set of high-reward behaviors already present in the pretrained distribution, effectively sacrificing entropy and diversity. This limits exploration and prevents the discovery of alternative strategies that could expand the base model's capabilities. Empirically, this effect is captured by the pass@$n$ metric, which measures the probability that at least one out of $n$ independently sampled rollouts succeeds. When $n$ is large, RL-fine-tuned models often underperform their pretrained counterparts because the latter retain greater diversity (Yue et al., 2025; Wu et al., 2025). Such diversity is practically important; it supports generalization to new tasks (Kumar et al., 2020) and amplifies test-time compute scaling (Snell et al., 2024).

The goal of this paper is to study how to induce policies to explore and refine a diverse repertoire of generations through RLFT. Our key insight is that algorithms should optimize objectives that explicitly encourage exploration and refinement of the diverse generations already embedded in the pretrained distribution. Standard regularization techniques, such as entropy bonuses, often induce local or token-level variation but fail to promote semantic or trajectory-level exploration and can be overshadowed by the RL objective. In contrast, we propose a unified formulation that directly

---

\*Equal contribution. Correspondence to {jubayer, ifdi1101}@stanford.edu.

optimizes for a diverse set of successful behaviors, encouraging the policy to generate broad, varied trajectories rather than collapsing onto a few high-reward ones.

To this end, we propose the framework of set reinforcement learning (set RL), where the objective is defined over a set of trajectories sampled independently and evaluated by a multi-sample objective (Tang et al., 2025). Unlike standard RL, which maximizes the likelihood of a single optimal trajectory, set RL maximizes the likelihood of an optimal set of trajectories sampled independently under a set-level objective. Within this framework, we introduce polychromic objectives which combine reward and diversity by scoring sets highly only if they contain both successful and diverse trajectories. Optimizing a policy with respect to this objective is a principled approach towards encouraging the policy to explore and search for a diverse set of generations that also maximize reward. We then instantiate one such objective and show how proximal policy optimization (PPO) (Schulman et al., 2017b) can be adapted to optimize it effectively, yielding a practical algorithm we call polychromic PPO. We evaluate our method on BabyAI (Chevalier-Boisvert et al., 2019), Minigrid (Chevalier-Boisvert et al., 2023), and Algorithmic Creativity (Nagarajan et al., 2025). Our results show that polychromic PPO achieves higher rewards and success rates, generates diverse trajectories that substantially improve pass@$n$ coverage, and generalizes more robustly to perturbations in the initial state.

## 2 PRELIMINARIES

We consider a Markov decision process (MDP) defined by state space $\mathcal{S}$, action space $\mathcal{A}$, transition dynamics distribution $p(s_{t+1} \mid s_t, a_t)$, reward function $r : \mathcal{S} \times \mathcal{A} \to \mathbb{R}$, initial state distribution $\rho_0$ and discount factor $\gamma \in (0, 1)$. In reinforcement learning (RL), the goal is to learn a policy that maximizes the value $V(\pi_\theta) = \mathbb{E}_{\tau \sim \pi_\theta} \left[ \sum_{t=0}^{\infty} \gamma^t r(s_t, a_t) \right] = \mathbb{E}_{\tau \sim \pi_\theta} \left[ R(\tau) \right]$ where $R(\tau)$ is the (discounted) sum of rewards in trajectory $\tau$. One widely used RL algorithm is proximal policy optimization (PPO) (Schulman et al., 2017b) which, iteratively, collects rollouts under a behavior policy $\pi_\beta$ and updates a policy $\pi_\theta$ by constraining the divergence between the two policies from growing too large: letting $r_t = \pi_\theta(a_t \mid s_t)/\pi_\beta(a_t \mid s_t)$, PPO optimizes an empirical estimate of the following:

$$\mathbb{E}_{s_t \sim d^{\pi_\beta}(\cdot), a_t \sim \pi_\beta(\cdot \mid s_t)} \left[ \min \left( r_t \hat{A}(s_t, a_t), \mathrm{clip}(r_t, 1 - \epsilon, 1 + \epsilon) \hat{A}(s_t, a_t) \right) \right]. \tag{1}$$

where $d^{\pi_\beta}(s)$ is the stationary state-visitation distribution under policy $\pi_\beta$.

## 3 REINFORCING EXPLORATION DURING RLFT

We aim to address the problem of entropy collapse during RLFT through a method that explicitly induces exploration by encouraging the generation of diverse trajectories. In §3.1, we introduce a variant of RL that allows for objectives beyond reward maximization and observe its various properties. This framework will provide us with a way to optimize objectives that are beyond return maximization, such as objectives that also encourage exploration. In §3.2, we specify the objective used within this framework for that purpose, which we call a polychromic objective. In §3.3, we propose our practical algorithm for optimizing the objective.

### 3.1 SET REINFORCEMENT LEARNING

We introduce a variant of the standard RL setup in which, given an objective function, we optimize over a *set* of trajectories. We call this framework **set reinforcement learning** (set RL), where the goal is to solve:

$$\max_\theta \mathbb{E}_{\tau_{1:n} \sim \pi_\theta(\cdot \mid s_0)} \left[ f(s_0, \tau_1, \cdots, \tau_n) \right]. \tag{2}$$

Here $f(s_0, \tau_1, \cdots, \tau_n)$ is some objective function over trajectories $\tau_{1:n} = \{\tau_1, \cdots, \tau_n\}$ sampled independently from the policy. This is in contrast to standard RL where the problem, $\max_\theta \mathbb{E}_{\tau \sim \pi_\theta(\cdot \mid s_0)} \left[ R(\tau) \right]$, uses an objective function, $R(\tau)$, defined over a single trajectory. The generality of set RL makes it a powerful tool for objectives beyond sum of rewards. Set RL objectives can be optimized using policy gradient methods by noting that

$$\nabla_\theta \mathbb{E}_{\tau_{1:n} \sim \pi_\theta(\cdot \mid s_0)}[f(s_0, \tau_{1:n})] = \mathbb{E}_{\tau_{1:n} \sim \pi_\theta(\cdot \mid s_0)} \left[ (f(s_0, \tau_{1:n}) - \hat{f}(s_0)) \sum_{i=1}^{n} \sum_{t=0}^{T} \nabla_\theta \log \pi_\theta(a_t^{(i)} \mid s_t^{(i)}) \right] \tag{3}$$

where $\hat{f}(s_0)$ is a variance-reduction baseline. In this paper, we use the baseline $\hat{f}(s_0) = \mathbb{E}_{\tau_{1:n} \sim \pi_\theta(\cdot|s_0)} [f(s_0, \tau_{1:n})]$. Note that the advantage term $f(s_0, \tau_{1:n}) - \hat{f}(s_0)$ is shared across all trajectories in the set $\tau_{1:n}$. This is a key defining feature of the set RL framework; the log-probability gradient of all trajectories in a set must be multiplied by the same factor. This contrasts with Tang et al. (2025), which also use $n$-sample objectives but employs trajectory-specific baselines (e.g., leave-one-out) leading to advantages of the form $f(s_0, \tau_{1:n}) - f(s_0, \tau_{1:i-1}, \tau_{i+1:n})$; so the update for trajectory $\tau_i$ depends on a baseline computed from the remaining trajectories, yielding individualized credit assignment. In our case, a uniform baseline provides a common update signal to all trajectories in the set, enabling optimization with respect to the quality of the entire *set* of trajectories. In other words, by definition, set RL does not distinguish between trajectories within a set, but instead optimizes the policy by comparing across sets as a whole. Note that the set RL framework can still be used to optimize the standard RL objective by choosing $f(s_0, \tau_{1:n}) = \frac{1}{n} \sum_{i=1}^{n} R(\tau_i)$. However, the framework allows for a broader class of objectives, such as inference-time objectives (Tang et al., 2025).

Notice that set RL is distinct from the framework of multi-objective reinforcement learning (Roijers et al., 2013). In multi-objective RL, we aim to maximize the same objective as in standard RL, except the rewards are vector-valued. As such, the objective is still defined over a single sampled trajectory, as opposed to being defined over a set of trajectories. While it is possible to consider vector-valued rewards in the set RL framework as well, in this paper we focus on optimizing scalar valued objectives of the form $f(s_0, \tau_{1:n}) \in \mathbb{R}$.

Before we move on to our proposed algorithm, it is helpful to construct a notion of value functions in the framework of set RL i.e., the expected sum of rewards as specified by the objective. We begin with a simplified (but impractical) setting. Suppose that at every state $s$ encountered during an on-policy rollout, we can sample a set of $n$ actions, $a_{1:n} \sim \pi_\theta(\cdot \mid s)$, which lead to $n$ trajectories stemming out of every state that branch even further along timesteps. Then, given this set of actions, $a_{1:n}$, taken from the state $s$, the policy gets the set reward $f(s, a_{1:n})$ with respect to the objective function $f$. Although such a setup is impractical for long-horizon tasks, analyzing it will help us better understand what set RL algorithms should aim to achieve.

Under this assumption, the data collection process naturally generates a state-visitation tree. Beginning at the root state $s_0$, each visited state $s$ branches into $n$ children, one for each sampled action. At depth $t$, the tree therefore contains $n^t$ states, denoted by $s_t^{(1)}, \ldots, s_t^{(n^t)}$. For each state $s_t^{(i)}$, the corresponding set of $n$ sampled actions is written as $(a_t)_{1:n}^{(i)}$. Given this tree-structured rollout and assuming an infinite-horizon discounted return, we define the value functions associated with set RL as follows:

**Definition 3.1** *Given a policy $\pi$ generating a state-visitation tree and a set objective $f : \mathcal{S} \times \mathcal{A}^n \to \mathbb{R}$, the set value function $V_\pi^\sharp(s; f)$ and the set Q-function $Q_\pi^\sharp(s, a_{1:n}; f)$ are defined as*

$$V_\pi^\sharp(s; f) = \mathbb{E}_\pi \left[ \sum_{t=0}^{\infty} \sum_{i=1}^{n^t} \gamma^t f(s_t^{(i)}, (a_t)_{1:n}^{(i)}) \Big| s_0 = s \right], \tag{4}$$

$$Q_\pi^\sharp(s, a_{1:n}; f) = \mathbb{E}_\pi \left[ \sum_{t=0}^{\infty} \sum_{i=1}^{n^t} \gamma^t f(s_t^{(i)}, (a_t)_{1:n}^{(i)}) \Big|_{(a_0)_{1:n}=a_{1:n}}^{s_0=s,} \right] \tag{5}$$

We use the notation $V^\sharp$ and $Q^\sharp$ to distinguish it from the value function and $Q$-function in standard RL. Here, we assume that the discount factor $\gamma \in (0, \frac{1}{n})$ to ensure values remain bounded - the range is smaller since at we add the expected sum of rewards from $n$ actions stemming out of each state. Intuitively, $V_\pi^\sharp(s)$ is the expected discounted return of the entire state tree rooted at $s$, where the reward at each node is given by the set objective. In contrast, $Q_\pi^\sharp(s, a_{1:n})$ evaluates the expected return of the tree that begins at $s$ with the specific action set $a_{1:n}$. Note that, although we assumed sets at the action-level instead of at the trajectory-level, this setting is equivalent to the trajectory-level set RL as in eq. (2) under the objective function $F(s, \tau_{1:n^T}) := \sum_{t=0}^{T} \sum_{i=1}^{n^t} \gamma^t f(s_t^{(i)}, (a_t)_{1:n}^{(i)})$ in the finite-horizon setting (in the infinite horizon setting, we would have take the limit $T \to \infty$).

These definitions help us better understand the objective of set RL. In standard reinforcement learning, we want to learn the policy such that the expected return from a trajectory is maximized.

In set RL, we want to learn the policy that maximizes the expected return from a *tree* generated using our policy is maximized. Given these definitions, we have the following result which is an extension of the performance difference lemma (Kakade & Langford, 2002) to the set RL setting (see §D.1 for proof):

**Lemma 3.2** *Given any two policies $\pi_\theta$ and $\pi_\beta$ and a fixed initial state $s_0$, under any set objective function $f$,*

$$V_{\pi_\theta}^\sharp(s_0; f) - V_{\pi_\beta}^\sharp(s_0; f) = \frac{1}{1 - \gamma n} \mathbb{E}_{s \sim d_{\pi_\theta}^\sharp(\cdot), a_{1:n} \sim \pi_\theta(\cdot|s)} \left[ A_{\pi_\beta}^\sharp(s, a_{1:n}; f) \right].$$

Similar to the standard reinforcement learning setting, this result says that if we update the policy $\pi_\beta$ to $\pi_\theta$ such that, at all states visited by $\pi_\theta$, the advantage $A_{\pi_\beta}^\sharp(s, a_{1:n}; f) = Q_{\pi_\beta}^\sharp(s, a_{1:n}; f) - V_{\pi_\beta}^\sharp(s)$ of a set of actions taken by our new policy $\pi_\theta$ is positive, then we will get a policy that has strictly higher performance (as measured by its value). This suggests that many of the principles underlying methods like PPO can be extended to the set RL paradigm as well for policy improvement.

Having seen the generality of set RL, we now construct an objective function to be used within this framework that will allow us to induce exploration.

## 3.2 A Practical Polychromic Objective

Our central construction is the notion of polychromic objectives that are aimed towards training an agent to explore and learn a diverse set of behaviors. Intuitively, these are set objective functions $f_{\text{poly}} : \mathcal{S} \times \mathcal{T}^n \to \mathbb{R}$ that jointly capture (1) the success of a set of trajectories in terms of reward and (2) the degree to which the set exhibits exploration or diversity. While we later generalize this construction in §5, in this section we focus on the specific objective used in our algorithm and experiments:

$$f_{\text{poly}}(s, \tau_{1:n}) := \frac{1}{n} \sum_{i=1}^n R(\tau_i) d(s, \tau_{1:n}), \tag{6}$$

where $R(\tau_i)$ is the discounted sum of rewards in trajectory $\tau_i$, and $d(s, \tau_{1:n})$ is a function that quantifies the diversity of trajectories within the set. We require that both $R(\tau_i)$ and $d(s, \tau_{1:n})$ are normalized between 0 and 1. Because the set-RL gradient uses a shared advantage for all trajectories in a set, this objective increases the likelihood of successful behaviors *and* diverse exploratory trajectories. Unlike prior approaches, the shared advantage term amplifies exploratory trajectories that do not (yet) yield high rewards, pushing the policy to discover diverse strategies.

Various diversity metrics have been studied and incorporated in reward functions in prior works; examples include the Vendi Score (Friedman & Dieng, 2023) and classifier-guided diversity (Zhang et al., 2025; Li et al., 2025). Our algorithm is designed to be agnostic to the choice of metric: given any diversity function, we evaluate the success and diversity of a set of trajectories and optimize the policy to maximize both in sets.

## 3.3 Polychromic PPO

In this section, we present an algorithm for optimizing eq. (6) by modifying PPO, which is motivated by the extension of the performance difference lemma to the set RL framework (as shown in Lemma 3.2). We choose to modify PPO since it is a widely used algorithm for reinforcement learning fine-tuning (Ouyang et al., 2022; Stiennon et al., 2020) known for stability and greater sample-efficiency (Achiam, 2018). However, the modifications we propose to instantiate our algorithm can be used to modify other algorithms too, such as REINFORCE, in order to optimize eq. (6). Our approach differs from standard PPO in two key respects: the method using which we sample on-policy rollouts and the advantage function used in the update.

A direct implementation of the definition of set advantage functions as in Lemma 3.2 would require sampling $n$ actions from every visited state, leading to exponential data requirements. To avoid this, we instead rely on vine sampling (Schulman et al., 2017a; Kazemnejad et al., 2025) for on-policy data collection. In vine sampling, after collecting an initial set of rollouts, we select a subset $\{s_1, \ldots, s_p\}$ of the states visited, called rollout states. At each rollout state $s_i$, we generate $N$ additional rollouts (called vines) starting from $s_i$. This procedure ensures that we obtain multiple

states with independently sampled trajectory sets stemming out of them. The particular scheme we use is closely related to the vine sampling scheme in TRPO (Schulman et al., 2017a); details are deferred to §B.1.1. Our algorithm, however, is compatible with any vine-sampling method that guarantees sufficient vine coverage. Note that, since vine sampling requires the ability to reset the environment, our algorithm is only applicable to environments where such resets are possible.

Given access to sets of trajectories from each rollout state, we can estimate the polychromic advantage. At a rollout state $s_t$ from which we generated $N > n$ trajectories, we estimate the polychromic advantage as

$$A^{\sharp}(s_t, a_t; f_{\text{poly}}) = \frac{1}{n} \sum_{i=1}^{n} R(\tau_i) d(s_t, \tau_{1:n}) - \hat{V}^{\sharp}(s_t; f_{\text{poly}})$$

where $a_t \in \tau_i$ for some $i \in 1, \ldots, n$. Since PPO requires an advantage defined for individual actions, we assign to each action $a_t$ the advantage of the set $\tau_{1:n}$ that contains it. In other words, given a set of trajectories $\tau_{1:n}$, all actions taken from $s_t$ in this set receive the same update signal as desired in set RL. We use the following Monte Carlo estimate of the value baseline: $\hat{V}^{\sharp}(s_t; f_{\text{poly}}) = \frac{1}{M} \sum_{i=1}^{M} f_{\text{poly}}(s_t, \tau_{1:n}^{(i)})$, where $\tau_{1:n}^{(i)}, i \in \{1, \cdots, M\}$ denotes the $M$ independently sampled sets of $n$ trajectories starting from $s_t$. This unbiased estimate was sufficient for our experiments, but one can trade off variance further by using biased estimates which we leave to future work.

For non-rollout states, the update remains the same as standard PPO, using generalized advantage estimation (GAE) (Schulman et al., 2018). As is often used in practical implementations of PPO, we also include a per-state KL penalty $D_{\text{KL}}(\pi_\beta(\cdot \mid s)|\pi_\theta(\cdot \mid s))$ at every state visited, which we found helpful for stability. The pseudocode is presented in algorithm 1, with modifications relative to PPO highlighted; extended pseudocode and implementation details are given in appendix B.

---

**Algorithm 1** Polychromic PPO

---

1: **for** iteration $= 1, 2, \ldots$ **do**
2:     Collect trajectories under $\pi_\beta$; rollout vines $\tau_{1:M}$ from rollout states
3:     **if** $s_t$ rollout state **then**
4:         Form sets, $g_1, \ldots, g_N$ of $n$ trajectories from $s_t$
5:         Set $A(s_t, a_t) = f_{\text{poly}}(s_t, g_i) - \frac{1}{N}\sum_{j=1}^{N} f_{\text{poly}}(s_t, g_j)$ for $(s_t, a_t) \in g_i$
6:     **else**
7:         Compute $\hat{A}(s_t, a_t)$ via GAE
8:     **end if**
9:     Update $\pi_\theta$ for $K$ epochs on minibatches $\mathcal{B}$ by maximizing the PPO objective in eq. (1)
10:     Set $\pi_\beta \leftarrow \pi_\theta$
11: **end for**

---

## 4 EXPERIMENTAL EVALUATION

Our experiments aim to answer two questions: (1) How does polychromic PPO, a set RL algorithm that explicitly encourages diverse trajectory generation, affect performance? More specifically, does improving diversity come at a significant cost in accuracy and success rate? (2) Does polychromic PPO encourage the policy to explore and learn diverse behaviors? In particular, learning to solve a task through diverse generations should, ideally, increase the pass@$n$ performance i.e., the probability of succeeding at least once when given multiple attempts. (3) Does encouraging the policy to explore and maximize the diversity of generated trajectories help the policy to be more robust to perturbations in the state-visitation distribution? To address these questions, we evaluate on Minigrid (Chevalier-Boisvert et al., 2023), BabyAI (Chevalier-Boisvert et al., 2019), and Algorithmic Creativity (Nagarajan et al., 2025). We evaluate on these environments since they allow for a diverse set of solutions We provide full environment and implementation details in §B, and briefly describe the environments below.

Minigrid and BabyAI are grid-world platforms with multiple rooms populated with keys, balls, and distractors. The agent receives natural-language goals (e.g., *open a red door and then go to the ball on your left after placing the grey ball next to a door*). We pretrain our policy on expert demonstrations (Chevalier-Boisvert et al., 2023; 2019); we then fine-tune and evaluate on 50 fixed configurations (each configuration specifies a grid layout and mission pair). In the

| Environment | Pretrained policy | REINFORCE | REINFORCE w/ UCB | PPO | PPO w/ UCB | Poly-PPO (ours) | Poly-PPO w/ UCB (ours) |
|---|---|---|---|---|---|---|---|
| Goto | (0.246, 34.2) | (0.533, 73.0) | (0.538, 73.4) | (0.406, 46.2) | (0.428, 47.4) | **(0.575, 80.2)** | **(0.561, 76.2)** |
| Pickup | (0.141, 21.4) | (0.259, 39.8) | (0.391, 56.0) | (0.283, 33.4) | (0.243, 27.8) | **(0.452, 63.2)** | **(0.486, 65.6)** |
| Synthseq | (0.157, 20.2) | **(0.325, 45.4)** | **(0.361, 47.8)** | (0.277, 32.2) | (0.224, 26.2) | **(0.341, 47.0)** | **(0.317, 43.2)** |
| Bosslevel | (0.212, 20.6) | (0.266, 33.4) | (0.286, 36.4) | (0.336, 38.8) | (0.310, 35.8) | **(0.378, 45.2)** | **(0.379, 46.8)** |
| Four Rooms | (0.469, 70.4) | (0.639, 89.6) | **(0.672, 92.6)** | **(0.618, 89.2)** | (0.502, 78.6) | **(0.666, 92.4)** | **(0.667, 93.2)** |

Table 1: Performance on BabyAI and MiniGrid tasks. Each entry is reported as $(x, y)$, where $x$ is the average episodic reward and $y$ is the success rate (in %). Results are averaged over 100 rollouts across 50 configurations and 3 random seeds.

*triangle discovery* task in Algorithmic Creativity, an agent must output sequences of triangles from undirected graphs that are not revealed to the agent in context; the agent must learn the graph from recalling data it has already seen and through further interactions during RLFT. We pretrain on triangles and edges drawn from 10 graphs, and then fine-tune on 3 graphs.

We compare polychromic PPO (Poly-PPO) to REINFORCE with baseline (Williams, 1992) and standard PPO (Schulman et al., 2017b). We also compare with a UCB-style regularization (Azar et al., 2017) where we add $\lambda_{\text{UCB}} \cdot \min\{1, N(s, a)^{-\frac{1}{2}}\}$ to every advantage $\hat{A}(s, a)$. Here, $N(s, a)$ is the number of times action $a$ was sampled from state $s$ and $\lambda_{\text{UCB}}$ is a hyperparameter. For Poly-PPO, we define the diversity function $d(s, \tau_{1:n})$ to be the fraction of distinct trajectories in $\tau_{1:n}$; in Minigrid/BabyAI, two trajectories are called distinct if they visit different sets of rooms and, in Algorithmic Creativity, two trajectories are distinct if they visit different sets of nodes. In both cases, $d = 0$ if all trajectories visit the same set of rooms or nodes.

All of these are long-horizon, sparse reward settings. Since the pretrained policy struggles to solve several of these long-horizon tasks, the policy must strategically explore during RLFT and avoid collapsing onto behaviors that solve only a subset while failing to generalize to the rest.

### 4.1 How Does Polychromic PPO Affect Performance?

The results summarizing performance, in terms of average reward and success rate, in Minigrid and BabyAI are provided in table 1. We find that Poly-PPO consistently matches or outperforms the best baseline in reward and success. Adding the UCB bonus helps the baselines, REINFORCE and PPO, improve performance in some environments. We find that UCB is complementary to Poly-PPO as well - the bonus enables the agent to achieve higher performance in *Pickup* and *Bosslevel*.

The results on *Triangle Discovery* are shown in fig. 2, validity is the number of valid triangles constructed. Diversity is the number of unique valid triangles and creativity metric is defined as the percentage of generations that are unique valid triangles not present in pretraining data (Nagarajan et al., 2025). PPO substantially increases validity compared to the pretrained policy, but lower creativity and diversity. On the other hand, Poly-PPO achieves slightly lower validity than vanilla PPO, but the gap is modest and highlights the tradeoff it strikes between success and exploration which we discuss in the next subsection.

### 4.2 Does Polychromic PPO Encourage Diverse Generations?

Success rates do not adequately represent the coverage of tasks that the policy can solve. Since success rates are averaged across all configurations, a policy that overfits to a subset may achieve high reward there while failing elsewhere. To probe this, we examine pass@$k$ curves - for each configuration, we provide the policy $k$ attempts and find the fractal of configurations the policy can solve, which is called the pass rate. As $k$ increases, methods that generate diverse trajectories should, ideally, achieve greater coverage/pass rate.

Pass@$k$ results on the BabyAI environments are shown in fig. 1. We first discuss all methods without the UCB bonus. We observe that REINFORCE does not improve pass rate sufficiently fast as the number of attempts increases: despite higher success rates overall, its coverage is lower than the pretrained policy at large $k$. PPO, on the other hand, starts off from a much lower pass rate than other methods at small $k$ but the pass rate increases as $k$ grows; however, it is still lower than the pretrained policy and significantly lower than Poly-PPO. This indicates that these baseline methods suffer from an inherent trade-off between diversity and accuracy in the generations. In comparison, Poly-PPO achieves substantially higher pass rate than all baselines. It also achieves

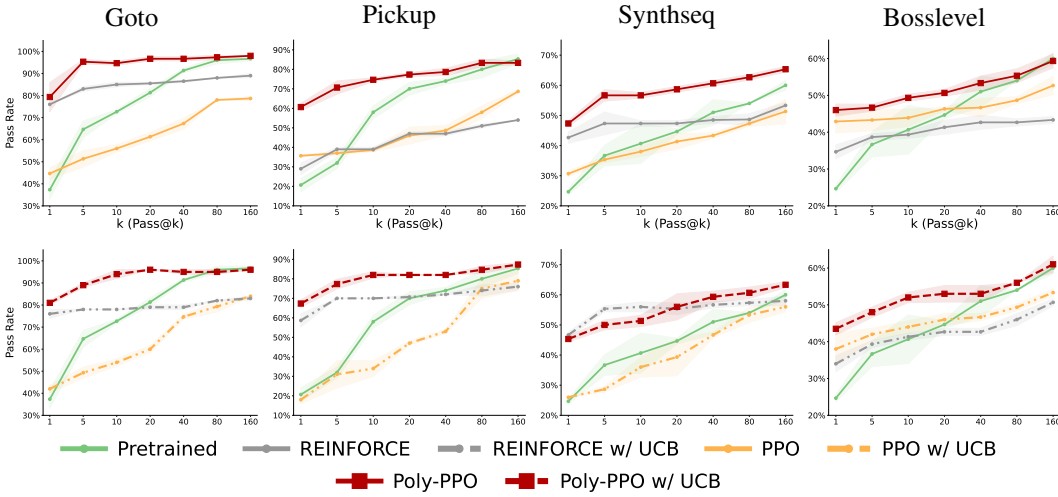

Figure 1: Pass@$k$ on BabyAI tasks. Top: methods without UCB. Bottom: methods with UCB. Each curve is pass rate vs. number of attempts.

equal or higher pass rate than the pretrained policy at almost all values of $k$. Another indicator for the higher diversity in generations is that the pass rate for Poly-PPO continues to rise until about $k = 80$, whereas the baselines saturate much earlier (around $k = 20$). The effect is pronounced, especially, in *Bosslevel* where Poly-PPO achieves around 15% higher pass rate as $k$ grows to 160 whereas baselines see modest increase.

We next examine the effect of adding the UCB bonus. For REINFORCE, the bonus improves pass rate at small $k$ in *Pickup* and *Synthseq*, but the gain saturates around $k = 10$ and vanishes for larger $k$; it has no effect in *Bosslevel* or *Goto*. For PPO, the bonus reduces pass rate at small $k$, and although it improves performance at larger $k$, the gap to the pretrained policy and Poly-PPO remains. By contrast, combining UCB with Poly-PPO yields equal or higher coverage across most $k$ (except small $k$ in *Synthseq*), showing that Poly-PPO preserves and refines pretrained diversity rather than collapsing onto narrow behaviors.

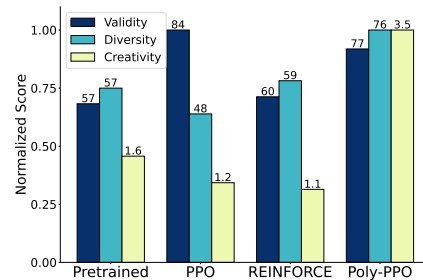

Figure 2: Results on Algorithmic Creativity. Bars show normalized values for each metric, with raw values above each bar.

In the *Triangle Discovery* task, We find that polychromic PPO achieves substantially higher diversity and creativity. In particular, Poly-PPO outperforms all baselines, including the pretrained policy, on both diversity and creativity metrics, as shown in fig. 2. Although it achieves slightly lower validity than PPO (significantly larger than REINFORCE though), Poly-PPO encourages broad exploration and the discovery of novel solutions. This trend is further reflected in the pass@$k$ evaluation (see fig. 3). Specifically, validity pass@$k$ measures whether at least one of the $k$ attempts forms a valid triangle; diff@$k$ quantifies the number of unique triangles obtained in $k$ attempts; and creativity pass@$k$ assesses whether at least one of the $k$ attempts is creative. We find that Poly-PPO outperforms baselines in both creativity and diversity metrics, while attaining greater validity performance than the pretrained policy. Notably, even though REINFORCE achieves high diversity, it comes at the significant cost in validity@1 where it is even below the pretrained policy.

We find that Poly-PPO achieves substantially higher diversity and creativity in the triangle discovery task. As shown in fig. 2, it outperforms all baselines, including the pretrained policy, on both metrics, while maintaining competitive validity above the pretrained policy. This pattern holds in the pass@$k$ evaluation (see fig. 3). Overall, Poly-PPO surpasses baselines in creativity and diversity while attaining higher validity than the pretrained policy, REINFORCE and nearly the same as PPO. Notably, although REINFORCE achieves high diversity, it does so at a steep cost: its validity@1 falls even below that of the pretrained policy.

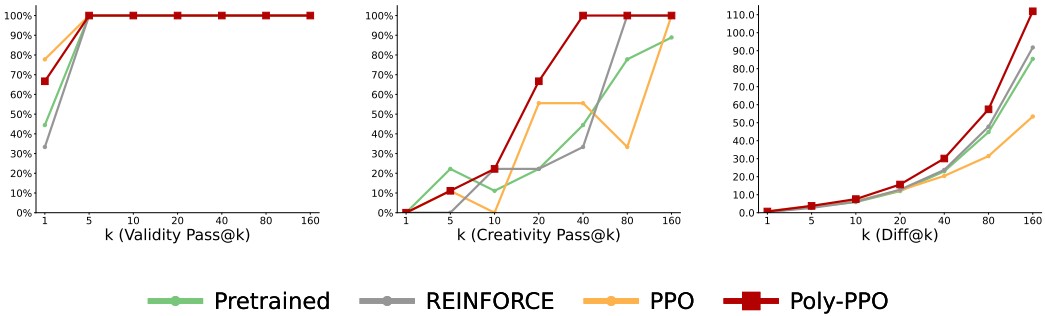

Figure 3: Pass@$k$ results on Algorithmic Creativity. For validity pass@$k$ and creativity pass@$k$, the agent gets a pass if at least one of the $k$ attempts was a valid and creative triangle, respectively. In diff@$k$ evaluation, we evaluate the number of generations that were unique given $k$ attempts.

| Environment | Pretrained policy | REINFORCE w/ Baseline | REINFORCE w/ UCB | PPO | PPO w/ UCB | Polychromic PPO | Poly-PPO w/ UCB |
|---|---|---|---|---|---|---|---|
| Goto | 30.2 | 41.3 | 37.1 | 21.1 | 18.4 | **60.6** | 54.3 |
| Pickup | 15.2 | 22.0 | 20.5 | 12.5 | 8.87 | **33.4** | 28.0 |
| Synthseq | 20.0 | 19.3 | 26.2 | 16.6 | 11.5 | **30.6** | 32.1 |
| Bosslevel | 23.8 | 22.5 | 27.6 | 26.6 | 28.2 | **34.3** | 32.8 |
| Four Rooms | 65.0 | 82.7 | 81.5 | 15.3 | 14.2 | **88.7** | 87.2 |

Table 2: Average pass rate (%) in one attempt on BabyAI and Minigrid tasks under large initial-state perturbations.

### 4.3 DOES POLYCHROMIC PPO GENERALIZE TO STATE PERTURBATIONS?

We evaluate generalization under perturbed initial states in Minigrid and BabyAI. For each grid–mission configuration, we first find all the rooms visited by the pretrained policy under high-temperature sampling over 100 rollouts. Then, we select 10 states randomly from inside each room as our initial states. Effectively, this changes the initial state to a completely different room in such a manner that the task remains solvable from the new initial state. Note that this randomization substantially changes the task for the agent; as shown in fig. 5, with respect to the new initial state, a successful trajectory would require very different strategies. From the new start state, we evaluate using pass@1 for all states in each layout. As shown in table 2, consistently, Poly-PPO generalizes more reliably than baselines under these perturbations.

## 5 ENTROPY ANALYSIS

In this section, we analyze how the entropy of a policy evolves when trained to optimize the polychromic objective in eq. (6). Our guiding question is: Under the polychromic objective, on which actions is a policy most likely to collapse its probability mass? In our analysis, we restrict our attention to the bandit setting i.e., time horizon $H = 1$, with binary rewards. We assume a discrete action space, and that the diversity function $d(s, a_{1:n})$ equals the fraction of actions in $a_{1:n}$ that are distinct ($d = 0$ if the set is a singleton). We assume that our policy has a softmax parameterization. Before turning to the polychromic objective itself, we extend the entropy analysis of Cui et al. (2025) to the set RL setting. In standard reinforcement learning with policy gradient methods, Cui et al. (2025) showed that, at state $s$, after one update to the policy, the change in the entropy of the policy at $s$ can be approximated as:

$$\mathcal{H}(\pi_\theta^{k+1} \mid s) - \mathcal{H}(\pi_\theta^k \mid s) \approx \alpha \mathrm{Cov}(\log \pi_\theta^k(a \mid s), A(s, a)) \tag{7}$$

where $\alpha$ is the learning rate. This result formalizes the understanding that a policy collapses its entropy onto high-reward actions when there is a strong covariance between the policy's probability mass on an action and the advantage of the action.

Given any set objective $f : \mathcal{S} \times \mathcal{A}^n \to \mathbb{R}$, we ask: how does the entropy of the policy change after one step update (from $\pi_\theta^k$ to $\pi_\theta^{k+1}$) when learning under this set-RL framework? The following proposition characterizes the first-order change (proof in §5.1):

**Proposition 5.1** *Consider the set-RL setup at state $s$. After one update to the policy, the change in entropy, $\Delta = \mathcal{H}\left(\pi_\theta^{k+1} \mid s\right) - \mathcal{H}\left(\pi_\theta^k \mid s\right)$, is given by*

$$\Delta \approx -\alpha \mathrm{Cov}_{a_{1:n}}(\frac{1}{n} \sum_{i=1}^n \log \pi_\theta^k(a_i \mid s), \mathrm{Cov}_{a'_{1:n}}(f(s, a'_{1:n}), \sum_{i,j=1}^n \mathbb{1}\{a_i = a'_j\})),$$

*where both covariances are taken with respect to $\pi_\theta^k(\cdot \mid s)$ and $\alpha$ is the learning rate.*

This result provides a lens for understanding when and where entropy collapse occurs. Intuitively, suppose that for some reference set $a_{1:n}$ there is a strong covariance between (i) the overlap of $a_{1:n}$ with sampled sets and (ii) the value of the objective $f$. As the policy concentrates more probability mass on such sets, the entropy decreases. Conversely, when the covariance is strongly negative, the policy reallocates probability mass to sets with higher value under $f$, which increases entropy. Thus, the central question becomes: which sets $a_{1:n}$ are most prone to entropy collapse under the polychromic objective? Our analysis proceeds by introducing and studying the following key object:

**Definition 5.2** *The scaffold value of a set of actions, $a_{1:n}$, under a policy $\pi$ and a set-RL objective $f : \mathcal{S} \times \mathcal{A}^n \to \mathbb{R}$ is defined to be*

$$\Lambda_f(a_{1:n}; \pi) := \mathrm{Cov}_{a'_{1:n} \sim \pi(\cdot \mid s)}(\hat{f}(s, a'_{1:n}), \frac{1}{I(a_{1:n})} \sum_{i,j=1}^n \mathbb{1}\{a'_i = a_j\})$$

*where $I(a_{1:n})$ is the maximum size of the intersection of $a_{1:n}$ with any other set $a'_{1:n}$.*

The scaffold represents, for every action set $a_{1:n}$, a measure of the policy's propensity to collapse its entropy around that set. We illustrate this further through the following lemma which shows us how the scaffold value of an action set affects the change in the probability of a policy sampling the set(proof in §5.1)

**Lemma 5.3** *Consider any set of actions $a_{1:n} \in \mathcal{A}^n$. The change in the log probability of sampling this set of actions after one policy update using set RL can be written as the following first-order approximation:*

$$\log \pi_\theta^{k+1}(a_{1:n} \mid s) \approx \log \pi_\theta^k(a_{1n} \mid s) + \alpha \Lambda_f(a_{1:n}; \pi_\theta^k) - \alpha C(\theta^k)$$

*where $C(\theta^k)$ is a function independent of $a_{1:n}$.*

With this apparatus in hand, we next show that the polychromic objective rules out collapse onto homogeneous sets of actions (proof in §D.4):

**Proposition 5.4** *Consider the polychromic objective in eq. (6). For any homogenous set $a_{1:n} = \{a\}$ where $r(s, a) = 1$, there exists $\epsilon \in (0, 1)$ such that $\Lambda_{f_{\mathrm{poly}}}(a) < 0$ when $\pi_\theta(a \mid s) > \epsilon$. Furthermore, the scaffold values of these homogenous sets satisfy the bound $\Lambda_{f_{\mathrm{poly}}}(a) \leq \sqrt{\frac{p(1-p)}{n}}$.*

This result shows that once a successful action $a$ accumulates sufficient probability mass, the polychromic objective automatically prevents further entropy collapse onto sets that only contain this action and, when we use larger sets in the set RL framework, the maximum scaffold value of a homogeneous set decreases. Before that threshold, the scaffold value of homogeneous sets of this action may be positive, but it is tightly bounded with the bound decreasing further as $n$ grows. This is desirable since it suggests that our policy learns to generate this action without incessantly collapsing probability mass on and memorizing such homogeneous sets. Next, we analyze the scaffold values of heterogenous sets with successful actions (proof in §D.5):

**Proposition 5.5** *Suppose $a_{1:n}$ is heterogeneous where each $a_i$ is unique with probability $\frac{p}{n}$ where $p \in (0, \frac{1}{n})$. Suppose exactly $q$ of the $n$ actions satisfy $r(s, a_i) = 1$, and that any other action $a' \notin a_{1:n}$ with $\pi_\theta(a' \mid s) > 0$ yields $r(s, a') = 0$. Then, the scaffold value of $a_{1:n}$ satisfies $\Lambda_{f_{\mathrm{poly}}}(a_{1:n}) > \frac{qp^n(1-p)}{n} \left(1 - \left(1 - \frac{1}{n}\right)^n\right)$.*

Note that the set in this proposition includes unsuccessful actions as well that contribute diversity. As such, there are, likely, several such heterogeneous sets with positive scaffold values that attract more probability mass than homogeneous sets. Moreover, lower bound guarantee increases as the number of successful actions in the set increases. The polychromic objective therefore channels entropy collapse toward those sets that balance success and exploration, rather than permitting collapse onto homogeneous behaviors. Together, these results motivate our construction of polychromic objectives in general. Broadly, such objectives reward both (i) the returns achieved by a set of trajectories and (ii) the diversity of the trajectories.

**Definition 5.6** *A polychromic objective is a function $\varphi : \mathcal{S} \times \mathcal{T}^n \to \mathbb{R}$ that factors as $\varphi(s, \tau_{1:n}) = \varphi^{(R)}(s, \tau_{1:n})\varphi^{(d)}(s, \tau_{1:n})$, where $\varphi^{(R)}$ and $\varphi^{(d)}$ are scalar-valued functions satisfying:*

1. *$\text{Cov}_{\tau_{1:n} \sim \pi_\theta(\cdot|s)}(\varphi^{(R)}(s, \tau_{1:n}), \sum_{i=1}^n R(\tau_i)) > 0$,*

2. *$\text{Cov}_{\tau_{1:n} \sim \pi_\theta(\cdot|s)}\left(\varphi^{(d)}(s, \tau_{1:n}), \sum_{i=1}^n \mathbb{1}\{\tau_i = \tau\}\right) < 0$ for any $\tau$, and*

3. *$\varphi^{(R)}(s, \cdot)$ and $\varphi^{(d)}(s, \cdot)$ share the same range, i.e.,*

$$\inf \varphi^{(R)}(s, \cdot) = \inf \varphi^{(d)}(s, \cdot), \quad \sup \varphi^{(R)}(s, \cdot) = \sup \varphi^{(d)}(s, \cdot),$$

*where the infimum and supremum are taken over all sets $\tau_{1:n}$.*

In other words, a polychromic objective factors into a reward component with positive covariance to return and a diversity component with negative covariance to homogeneity. Their product enforces that both success and diversity are indispensable.

## 6 RELATED WORK

RL fine-tuning often suffers from entropy collapse, where policies concentrate on a few high-reward behaviors and lose coverage of the pretrained distribution (Cui et al., 2025; Wu et al., 2025; Yue et al., 2025). Entropy bonuses (Haarnoja et al., 2017; 2018; Schulman et al., 2017b; Seo et al., 2021; Islam et al., 2019) mitigate this locally but may struggle to induce semantic or trajectory-level exploration. Ecoffet et al. (2021) aims to explore more from promising states and trains a policy to robustly handle stochasticity. Other approaches, more similar to our work, explicitly reward diversity, e.g., diversity-weighted objectives (Li et al., 2025), UCB bonuses (He et al., 2025; Lanchantin et al., 2025), batch-level exploration rewards (Song et al., 2025) or use covariance controls (Cui et al., 2025). The crucial distinction our work makes is the set RL framework that enables learning a set of behaviors wherein some trajectories receive positive gradient updates for exploration despite not contributing the maximum rewards attained in the set.

## 7 CONCLUSION

We framed the problem of learning a diverse set of successful behaviors in terms of set reinforcement learning and proposed optimizing the polychromic objective, which evaluates sets of actions using both reward and diversity. We derived polychromic-PPO, a variant of PPO that incorporates vine sampling and a modified advantage estimator to optimize this objective. There are several limitations of our approach. Firstly, our approach requires the ability to reset the environment to any set to enable vine sampling. Furthermore, ensuring sufficient vine coverage can be computationally demanding. Future work could develop more efficient, low-variance gradient estimators, or adopt curriculum and annealing schemes that balance exploration early in training with exploitation later.

## 8 ACKNOWLEDGMENTS

We would like to thank Amber Xie, Joey Hejna, Suvir Mirchandani, Hung-Chieh Fang and Andy Tang for feedback on an earlier version of the paper. We thank Ayush Chakravarthy for helpful comments on the experimental evaluations. We thank Yoonho Lee for several valuable comments on formalizing diversity. We thank Hamidur Rahman and Pervin Akhter for helpful guidance on the formalization of the problem and feedback on the project in general. We thank Zabed Hasan and Fazilat Afreen for their insightful feedback on experimental evaluations. This work is supported by an NSF

CAREER award, NSF award # 1941722, ONR YIP, DARPA YFA award # W911NF2210214, and Schmidt Sciences.

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

## A LLM Use

LLMs were used to help with polishing the paper and for coding assistance in low-level implementation. We take full responsibility for all ideas, experiments, results, and conclusions presented in this paper.

## B Implementation Details

**BabyAI and MiniGrid.** For BabyAI tasks, the policy conditions on the grid image, the agent's direction embedding, and the mission text (encoded by a GRU); the action space is the standard BabyAI/MiniGrid discrete set *left, right, forward, pickup, drop, toggle, done*. We provide example configurations for each task in fig. 4. We train a CNN–GRU policy that outputs action logits. In *MiniGrid–FourRooms*, the action space is identical, but the observation excludes a mission (as the goal specification is fixed), so we use a compact MLP that produces action logits conditioned on a flattened image observation. During pretraining, we use an 80/20 train–test split of expert demonstrations for each task, except for *Synthseq* and *BossLevel*, where we found that the full dataset was required to obtain a reasonably strong base policy. We used the dataset from Younis et al. (2024). We pretrain by minimizing the cross-entropy loss with an entropy regularizer.

During RLFT, we fine-tune on 50 configurations. For all tasks except *BossLevel* and *Synthseq*, these configurations are drawn from the pretraining test set; for *BossLevel* and *Synthseq*, we did not carve out a separate test set. If the agent reaches the goal at time $t$ (episode horizon $H=100$), it receives the reward $1 - 0.5 \cdot \frac{t}{H}$; otherwise $r = 0$. (BabyAI/MiniGrid defaults to $1 - 0.9 \cdot \frac{t}{H}$; we found that lowering the time penalty improved diversity during RLFT, especially for REINFORCE, by reducing the disadvantage of longer but successful trajectories.)

**Algorithmic Creativity (Triangle Discovery).** In *Triangle Discovery*, an agent must output a sequence of edge tokens that form a valid triangle in an *unobserved* undirected graph. The input sequence comprises a graph index plus a prefix prompt and the agent's past outputs. We set the maximum sequence length to be 11. The action space is a discrete vocabulary of size 1017. We pretrain a decoder-only Transformer with masked cross-entropy for 25 epochs. The pretraining dataset, from Nagarajan et al. (2025), contains 15,000 samples per graph, with the same being a triangle with probability $\frac{1}{3}$ and an edge with probability $\frac{2}{3}$ edges. Each graph has 999 nodes. For RLFT, we fine-tune on 3 fixed graphs. The reward is sparse: $+1$ for a valid triangle, 0 otherwise.

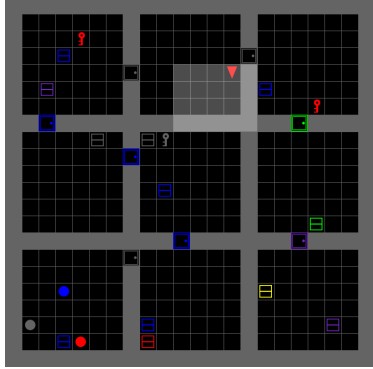

(a) *GoTo:* mission "go to purple box."

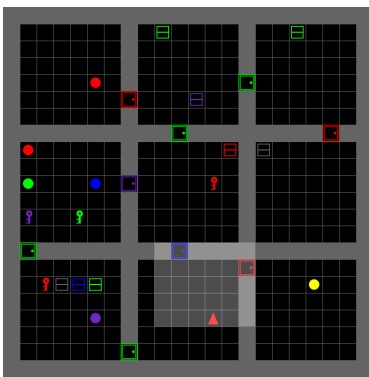

(b) *Pickup:* mission "pick up a green ball."

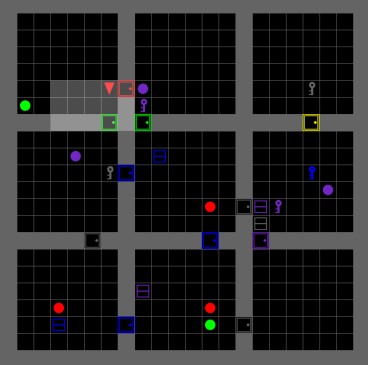

(c) *Synthseq:* mission "put the ball on your right next to a red ball and pick up a purple ball after you open a grey door."

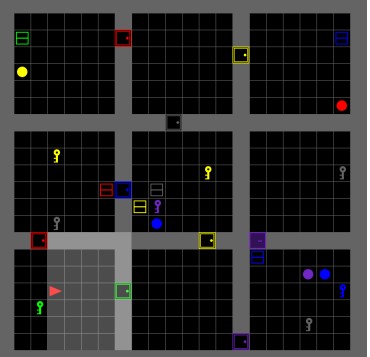

(d) *Bosslevel:* mission "put a yellow key next to the blue key and pick up a ball after you pick up a yellow key."

Figure 4: Example BabyAI environments and their missions.

## B.1 POLYCHROMIC PPO

We summarize implementation details for polychromic PPO , beginning with the *vine sampling* scheme used for on-policy data collection, then additional stability techniques and full pseudocode in algorithm 2.

### B.1.1 VINE SAMPLING

In this section, we describe the vine sampling scheme (Schulman et al., 2017a) we used for polychromic PPO . In vine sampling, we first generate a number of trajectories. Then, we select a subset of the states visited, denoted by $\{s_1, \ldots, s_p\}$ - this is called the *rollout set*. For each state $s_i$ in the rollout set (we call this a *rollout state*), we generate a set of trajectories $\tau_{1:N} \sim \pi_\beta(\cdot \mid s_i)$. To avoid notational overload and to make sample accounting explicit, we distinguish:

- $n$: the set size used by the set-RL objective (number of trajectories per set)
- $N$: the number of *rollouts* collected from each rollout state/vine state
- $p$: the number of *rollout states* selected along each seed rollout
- $M$: the trajectory budget i.e., maximum number of trajectories we can collect.

We now discuss the method we used to select the set of rollout states. Our goal is to identify multiple states from which we want to generate vines so that we can use the polychromic advantage to update the policy at a large number of states. Suppose, we have a trajectory budget $M$, i.e., we are allowed to generate at most $M$ trajectories during the on-policy data collection. We first sample $N$ rollouts independently where $N > n$. Now, for each of these $N$ trajectories, we identify $p$ rollout states according to some criterion. Some examples of rollout state criterion are: (1) Top $p$ states with the

highest entropy; sample more trajectories from states where the policy is more uncertain, (2) Top $p$ states with the highest critic losses; sample more trajectories where our critic is wrong/biased, and (3) $p$ equally spaced out states. Suppose the main trajectory is $T$ timesteps long. We select the states at timestep $\frac{T}{p+1}, \frac{2T}{p+1}, \cdots, \frac{pT}{p+1}$.

In this paper, we used the third criterion. Note that, in this scheme, we generate, in total, $N+N^2(p-1)$ trajectories. Since we are allowed to sample at most $M$ trajectories, we must select $N$ and $p$ such that $N > n$ and $N + N^2(p-1) \leq M$. Across all environments, we set a trajectory budget $M = 136$ for all methods. As such, we use sets of size $n = 4$ for set RL, $N = 8$ vines at each rollout state, and $p = 2$ rollout states per trajectory. Note that at each of these rollout states, given we generate $N$ rollouts, we can find $\binom{N}{n}$ sets for our set RL algorithm. For our chosen hyperparameters, this was sufficiently large number of sets from each rollout state allowing us to use several sets to compute the baseline.

### B.1.2 OTHER IMPLEMENTATION DETAILS AND PSEUDOCODE

Now, we will discuss other implementation details for polychromic PPO . We provide the pseudocode for the complete algorithm in algorithm 2.

We found that adding the KL penalty from the behavior policy was helpful for stability. In the absence of it, in some tasks, the model's performance collapses after a certain number of training epochs. This is likely because our method explicitly encourages exploration and the KL penalty provides an anchor that prevents the model from drifting too far.

In practical implementation, we found that adding a window within which we update all the advantages to the polychromic advantage to be useful. As shown in algorithm 2, we do not only set the advantage at the rollout state $s_t$ to be the polychromic advantage - instead, we set the advantages at states $s_t, s_{t+1}, \cdots, s_{t+W}$ to be the polychromic advantage even though we do not generate vines from $s_{t+1}, \cdots, s_{t+W}$. This ensures that the exploratory behavior that the polychromic advantage encourages is induced at all states throughout the window. Otherwise, although the policy might be exploratory at $s_t$, it might revert to being purely exploitative at all subsequent states due to the updates using the standard PPO objective; this would cause the policy to not explore. This issue becomes pronounced in environments where, despite being exploratory at $s_t$, the exploitative behavior in all subsequent states may override the exploration in $s_t$. This is why this implementation trick was important in BabyAI and Minigrid while, in Algorithmic Creativity, we set $W = 0$ since once the policy visits a diverse set of nodes from $s_t$, it is not easy to merge paths.

| Hyperparameter | Value |
|---|---|
| PPO epochs | 2 |
| Minibatch size | 64 |
| Discount ($\gamma$) | 1.0 |
| GAE parameter ($\lambda$) | 0.95 |
| Clipping parameter ($\epsilon$) | 0.2 |
| Actor learning rate | $1 \times 10^{-5}$ |
| Critic learning rate | $1 \times 10^{-4}$ |
| Value loss coefficient ($c_v$) | 0.5 |
| KL coefficient ($\beta_{\mathrm{KL}}$) | $\{0.005, 0.01, 0.05, 0.1\}$ |
| Max grad norm | 0.5 |
| Temperature | 1.0 |
| Size of sets ($n$) | 4 |
| Num. vines at state ($N$) | 8 |
| Polychrome window ($W$) | 5 for BabyAI/Minigrid, 0 for Algorithmic Creativity |
| Num. vine states ($p$) | 2 |

Table 3: Polychromic PPO hyperparameters. All hyperparameters are fixed apart from the KL coefficient $\beta_{\mathrm{KL}}$ for which we provide the set we sweep over.

---

**Algorithm 2** Polychromic PPO

---

**Require:** pretrained Policy $\pi_\beta$, value function $V_\phi$, discount $\gamma$, GAE parameter $\lambda$, clipping $\epsilon$, policy epochs $K$, value coef $c_v$, KL target coef $\beta_{\text{KL}}$
  (Poly-PPO specific:) number of sets $N$, set size $n$, number of vine states $p$, number of vines $N$, window length $W$.
 1: Initialize $\pi_\theta \leftarrow \pi_\beta$
 2: **for** iteration $= 1, 2, \ldots$ **do**
 3:     **Collect on-policy data using fractal sampling:**
 4:         Initialize `rollout_states` $\leftarrow \{\}$           $\triangleright$ dictionary: state $\mapsto$ list of trajectories
 5:         Roll out $N$ trajectories using $\pi_\beta$
 6:         **for** each trajectory $\tau$ **do**
 7:             select $p$ vine states from $\tau$
 8:             **for** each vine state $s$ **do**
 9:                 Roll out $N$ trajectories from $s$ using $\pi_\beta$
10:                 Append the new trajectories to `rollout_states`$[s]$
11:             **end for**
12:         **end for**
13:     **Compute advantages:**
14:     **if** $s \notin$ `rollout_states` **then**
15:         $\delta_t \leftarrow r_t + \gamma V_\phi(s_{t+1}) - V_\phi(s_t)$
16:         $A_t \leftarrow \text{GAE}(\delta_{t:T}, \gamma, \lambda)$           $\triangleright$ generalized advantage estimation
17:         $\hat{R}_t \leftarrow A_t + V_\phi(s_t)$
18:     **else if** $s \in$ `rollout_states` **then**
19:         Create groups $g_1, g_1, \cdots, g_M$ of $n$ trajectories from `rollout_states`$[s]$.
20:         Compute set scores $\text{score}(g_i) = f_{\text{poly}}(s, \tau_{1:n})$ for $\tau_{1:n} \in g_i$ (eq. (6))
21:         Compute baseline $\hat{f}(s) = \frac{1}{M} \sum_{i=1}^{M} \text{score}(g_i)$.
22:         Define polychromic advantage of pairs $(s_t, a_t), \ldots, (s_{t+W}, a_{t+W}) \in \tau$ for each $\tau \in g_i$:

$$A^{\text{poly}}(s_{t'}, a_{t'}) \leftarrow \text{score}(g_i) - \hat{f}(s) \qquad \triangleright \text{ polychromic advantage}$$

23:     **end if**
24:     Normalize $\{A_t\}$
25:     **for** epoch $= 1, \ldots, K$ **do**
26:         **for** minibatch $\mathcal{B}$ **do**
27:             Compute ratios $r_t(\theta) \leftarrow \frac{\pi_\theta(a_t|s_t)}{\pi_\beta(a_t|s_t)}$
28:             Policy loss:

$$\mathcal{L}_\pi(\theta) \;=\; -\frac{1}{|\mathcal{B}|} \sum_{t \in \mathcal{B}} \min\Big( r_t(\theta) A_t,\; \text{clip}(r_t(\theta), 1 - \epsilon, 1 + \epsilon)\, A_t \Big)$$

29:             Value loss: $\mathcal{L}_V(\phi) = \frac{1}{|\mathcal{B}|} \sum_{t \in \mathcal{B}} \big( V_\phi(s_t) - \hat{R}_t \big)^2$
30:             KL penalty: $\mathcal{L}_{\text{KL}}(\theta) = \frac{1}{|\mathcal{B}|} \sum_{t \in \mathcal{B}} \text{KL}\big( \pi_\beta(\cdot|s_t) \,\|\, \pi_\theta(\cdot|s_t) \big)$
31:             Total loss:

$$\mathcal{L}(\theta, \phi) \;=\; \mathcal{L}_\pi(\theta) \;+\; c_v\, \mathcal{L}_V(\phi) \;+\; \beta_{\text{KL}}\, \mathcal{L}_{\text{KL}}(\theta)$$

32:             Take gradient step on $\theta, \phi$ to minimize $\mathcal{L}$
33:             Update $\pi_\beta \leftarrow \pi_\theta$
34:         **end for**
35:     **end for**
36: **end for**

---

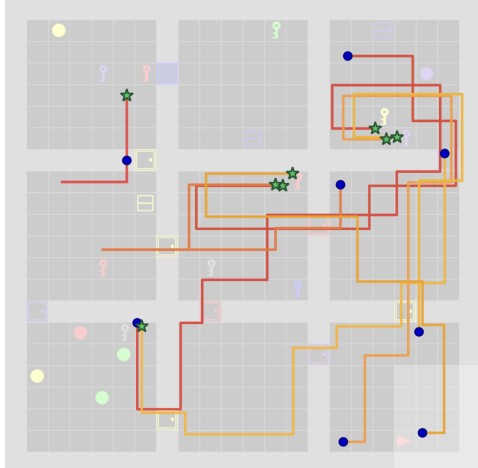

Figure 5: Generalization under state perturbations in BabyAI BossLevel environment. The figure shows successful rollouts across perturbed initial states (blue circles), highlighting diverse strategies learned by the agent.

## C   GENERALIZATION EXPERIMENT

To evaluate the agent's ability to generalize to initial state perturbations, we designed the following experiment. First, for each environment seed, we perform 100 rollouts with the pretrained policy to identify all reachable rooms. Then, for each unique room visited, we evaluate the target policy (fine-tuned using RL) by placing the agent at 10 randomly selected starting positions within that room. Performance is measured using the pass@1 metric, which calculates the success rate on the first attempt from these novel starting points. This randomization presents a significant challenge, as a successful trajectory from a new initial state often requires substantially different strategies than those effective from the original start state as shown in fig. 5.

# D  PROOFS

## D.1  PROOF OF LEMMA 3.2

Given the set reinforcement learning framework, we first consider the set up where, from each state $s$ visited during on-policy rollouts, we can generate $n$ actions, $a_{1:n}$. As such, from each state, we generate a tree where each node branches out into $n$ children. We denote all the nodes at depth $t$ of this tree by $s_t^{(1)}, \cdots, s_t^{(n^t)}$.

Furthermore, if from a state $s$, the agent generates the set of actions $a_{1:n}$, the agent gets a reward $f_{\text{poly}}(s, a_{1:n}) = \frac{1}{n} \sum_{i=1}^{n} r(s, a_i) d(s, a_{1:n})$ where $r(s, a_i)$ is the original reward function in the MDP and $d(s, a_{1:n})$ is some function measuring diversity.

For simplicity, we assume that the initial state is fixed. Note that we can construct the distribution of states visited by policy $\pi_\theta$ at time $t$ in this tree as follows:

$$
\begin{aligned}
&\mathbb{P}(s_0 \to \{s_t^{(1)}, \cdots, \{s_t^{(n^t)}\}, t, \pi_\theta) \\
&= \mathbb{P}(s_0 \to \{s_t^{(1:n^t)}\}, t, \pi_\theta) \\
&= \sum_{(a_0)_{1:n}} \pi_\theta((a_0)_{1:n} \mid s_0) \sum_{s_1^{(1:n)}} P(s_1^{(1:n)}) \mid s_0, (a_0)_{1:n}) \cdots \\
&\quad \times \sum_{(a_{t-1})_{1:n}^{(1)}, \cdots, (a_{t-1})_{1:n}^{(n^{t-1})}} \pi_\theta((a_{t-1})_{1:n}^{(1)}, \cdots, (a_{t-1})_{1:n}^{(n^{t-1})} \mid s_{t-1}^{(1)}, \cdots, s_{t-1}^{(n^{t-1})}) \\
&\quad \times P(s_t^{(1:n^t)} \mid s_{t-1}^{(1:n^{t-1})}, (a_{t-1})_{1:n}^{(1)}, \cdots, (a_{t-1})_{1:n}^{(n^{t-1})})
\end{aligned}
$$

Here, each $\pi_\theta(a_{1:n} \mid s) = \prod_{i=1}^{n} \pi_\theta(a_i \mid s)$. Similarly, we use the following shorthand:

$$
\pi_\theta((a_{t-1})_{1:n}^{(1)}, \cdots, (a_{t-1})_{1:n}^{(n^{t-1})} \mid s_{t-1}^{(1)}, \cdots, s_{t-1}^{(n^{t-1})}) = \prod_{i=1}^{n^{t-1}} \pi_\theta((a_{t-1})_{1:n}^{(i)} \mid s_{t-1}^{(i)}).
$$

Before we prove the lemma, we make the following observation: suppose the environment's state transition dynamics is deterministic, then the polychromic $Q$-function can be written using the polychromic value function as follows:

$$
Q_\pi^\sharp(s, a_{1:n}) = f_{\text{poly}}(s, a_{1:n}) + \gamma \sum_{i=1}^{n} V_\pi^\sharp(s_1^{(i)})
$$

where $s_1^{(1)}, \cdots, s_1^{(n)}$ are the states reached from $s$ after taking actions $a_{1:n}$ independently.

One can easily verify this by starting from the definition of the set $Q$-function and noting that:

$$
\mathbb{E}\left[ \sum_{t=1}^{\infty} \gamma^t \sum_{i=1}^{n^t} f_{\text{poly}}(s_t^{(i)}, (a_t)_{1:n}^{(i)}) \mid s_0 = s, a_{1:n}(s_0) = a_{1:n} \right]
$$

$$
= \mathbb{E}\left[ \gamma \sum_{i=1}^{n} f_{\text{poly}}(s_1^{(i)}, (a_1)_{1:n}^{(i)}) + \sum_{t=2}^{\infty} \gamma^t \sum_{i=1}^{n^t} f_{\text{poly}}(s_t^{(i)}, (a_t)_{1:n}^{(i)}) \Big| s_0 = s, (a_0)_{1:n} = a_{1:n} \right]
$$

$$
= \gamma \sum_{i=1}^{n} \mathbb{E}_{(a_1)_{1:n}^{(i)}} \left[ f_{\text{poly}}(s_1^{(i)}, (a_1)_{1:n}^{(i)}) + \mathbb{E}\left[ \sum_{t=2}^{\infty} \gamma^t \sum_{i=1}^{n^t} f_{\text{poly}}(s_t^{(i)}, (a_t)_{1:n}^{(i)}) \Big| s_0 = s, (a_0)_{1:n} = a_{1:n}, s_1^{(i)}, (a_1)_{1:n}^{(i)} \right] \right]
$$

$$
= \gamma \sum_{i=1}^{n} \mathbb{E}_{(a_1)_{1:n}^{(i)}} \left[ f_{\text{poly}}(s_1^{(i)}, (a_1)_{1:n}^{(i)}) + \gamma \sum_{i=1}^{n} \mathbb{E}\left[ \sum_{t=2}^{\infty} \gamma^{t-1} \sum_{j=1}^{n^{t-1}} f_{\text{poly}}(s_t^{(n(i-1)+j)}, (a_t)_{1:n}^{(n(i-1)+j)}) \Big|_{(a_0)_{1:n}=a_{1:n}, (a_1)_{1:n}^{(i)}}^{s_0=s, s_1^{(i)}} \right] \right]
$$

$$= \gamma \sum_{i=1}^{n} V_{\pi}^{\sharp}(s_1^{(i)})$$

Given this and our constructions of the polychromic value functions, the proof of Lemma 3.2 follows the same procedure as the standard performance difference lemma in Kakade & Langford (2002).

**Lemma D.1** *Given the state visitation tree generated by policy $\pi_\theta$ and any normalized objective function $f$,*

$$\mathbb{E}_{\pi_\theta}\left[\sum_{t=0}^{\infty}\gamma^t\sum_{i=1}^{n^t}f(s_t^{(i)},(a_t)_{1:n}^{(i)})\right] = \frac{1}{1-\gamma n}\mathbb{E}_{s\sim d_{\pi_\theta}^{\sharp}(s),a_{1:n}\sim\pi_\theta(\cdot|s)}\left[f(s,a_{1:n})\right] \tag{8}$$

*where*

$$d_{\pi}^{\sharp}(s) = (1-\gamma n)\sum_{t=0}^{\infty}\sum_{i=1}^{n}\gamma^t\mathbb{P}(s_0\to s,t,\pi_\theta).$$

*Proof.*

$$\mathbb{E}\left[\sum_{t=0}^{\infty}\sum_{i=1}^{n^t}\gamma^t f(s_t^{(i)},(a_t)_{1:n}^{(i)})\right]$$

$$= \sum_{t=0}^{\infty}\gamma^t\sum_{i=1}^{n^t}\mathbb{E}\left[f(s_t^{(i)},(a_t)_{1:n}^{(i)})\right]$$

$$= \sum_{t=0}^{\infty}\gamma^t\sum_{i=1}^{n^t}\sum_{s_t^{(i)}}\mathbb{P}(s_0\to s_{t,i},t,\pi_\theta)\mathbb{E}_{a_{1:n}(s_{t,i})}\left[f(s_t^{(i)},(a_t)_{1:n}^{(i)})\right]$$

$$= \frac{1}{1-\gamma n}(1-\gamma n)\sum_{t=0}^{\infty}\gamma^t\sum_{i=1}^{n^t}\sum_{s_t^{(i)}}\mathbb{P}(s_0\to s_t^{(i)},t,\pi_\theta)\mathbb{E}_{a_{1:n}(s_{t,i})}\left[f(s_t^{(i)},(a_t)_{1:n}^{(i)})\right]$$

$$= \frac{1}{1-\gamma n}\mathbb{E}_{s\sim d_{\pi_\theta}^{\sharp}(s),a_{1:n}\sim\pi_\theta(\cdot|s)}\left[f(s,a_{1:n})\right].$$

$\square$

**Lemma 3.2 (restated).** Given any two policies $\pi_\theta$ and $\pi_\beta$,

$$V_{\pi_\theta}^{\sharp}(s) - V_{\pi_\beta}^{\sharp}(s) = \frac{1}{1-\gamma n}\mathbb{E}_{s\sim d_{\pi_\theta}^{\sharp}(\cdot),a_{1:n}\sim\pi_\theta(\cdot|s)}\left[A_{\pi_\beta}^{\sharp}(s,a_{1:n})\right]. \tag{9}$$

*Proof.* We first, show that

$$V_{\pi_\theta}^{\sharp}(s) - V_{\pi_\beta}^{\sharp}(s) = \mathbb{E}\left[\sum_{t=0}^{\infty}\sum_{i=1}^{n^t}Q_{\pi_\beta}^{\sharp}(s_t^{(i)},(a_t)_{1:n}^{(i)}) - V_{\pi_\beta}^{\sharp}(s_t^{(i)}) \mid s_0 = s\right]. \tag{10}$$

We show this by the following simplification:

$$V_{\pi_\theta}^{\sharp}(s) - V_{\pi_\beta}^{\sharp}(s)$$

$$= \mathbb{E}_{\pi_\theta}\left[\sum_{t=0}^{\infty}\gamma^t\sum_{i=1}^{n^t}f_{\text{poly}}\left(s_t^{(i)},(a_t)_{1:n}^{(i)}\right) \mid s_0 = s\right] - V_{\pi_\beta}^{\sharp}(s)$$

$$= \mathbb{E}_{\pi_\theta}\left[\sum_{t=0}^{\infty}\gamma^t\sum_{i=1}^{n^t}f_{\text{poly}}\left(s_t^{(i)},(a_t)_{1:n}^{(i)}\right) \mid s_0 = s\right] - V_{\pi_\beta}^{\sharp}(s)$$

$$+ \mathbb{E}_{\pi_\theta} \left[ \sum_{t=0}^{\infty} \gamma^t \sum_{i=1}^{n^t} \gamma \sum_{j=1}^{n} V_{\pi_\beta}^\sharp (s_{t+1}^{(n(i-1)+j)}) \,\Big|\, s_0 = s \right]$$

$$- \mathbb{E}_{\pi_\theta} \left[ \sum_{t=0}^{\infty} \gamma^t \sum_{i=1}^{n^t} \gamma \sum_{j=1}^{n} V_{\pi_\beta}^\sharp (s_{t+1}^{(n(i-1)+j)}) \,\Big|\, s_0 = s \right]$$

$$= \mathbb{E}_{\pi_\theta} \left[ \sum_{t=0}^{\infty} \gamma^t \sum_{i=1}^{n^t} \Big( f_{\text{poly}} \big( s_t^{(i)}, (a_t)_{1:n}^{(i)} \big) + \gamma \sum_{j=1}^{n} V_{\pi_\beta}^\sharp (s_{t+1}^{(n(i-1)+j)}) \Big) \right]$$

$$- \mathbb{E}_{\pi_\theta} \left[ \sum_{t=0}^{\infty} \gamma^t \sum_{i=1}^{n^t} V_{\pi_\beta}^\sharp (s_t^{(i)}) \right].$$

Now, the first term on the right hand side can be simplified (we suppress the condition that $s_0 = s$ in terms of notation):

$$\mathbb{E}_{\pi_\theta} \left[ \sum_{t=0}^{\infty} \gamma^t \sum_{i=1}^{n^t} \Big( f_{\text{poly}} \big( s_t^{(i)}, (a_t)_{1:n}^{(i)} \big) + \gamma \sum_{j=1}^{n} V_{\pi_\beta}^\sharp (s_{t+1}^{(n(i-1)+j)}) \Big) \right]$$

$$= \sum_{t=0}^{\infty} \gamma^t \sum_{i=1}^{n^t} \mathbb{E}_{\pi_\theta} \left[ f_{\text{poly}} \big( s_t^{(i)}, (a_t)_{1:n}^{(i)} \big) + \gamma \sum_{j=1}^{n} V_{\pi_\beta}^\sharp (s_{t+1}^{(n(i-1)+j)}) \right]$$

$$= \sum_{t=0}^{\infty} \gamma^t \sum_{i=1}^{n^t} \mathbb{E}_{s_{t,i},(a_t)_{1:n}^{(i)}} \left[ \mathbb{E} \left[ f_{\text{poly}} \big( s_t^{(i)}, (a_t)_{1:n}^{(i)} \big) + \gamma \sum_{j=1}^{n} V_{\pi_\beta}^\sharp (s_{t+1}^{(n(i-1)+j)}) \,\Big|\, s_{t,i}, (a_t)_{1:n}^{(i)} \right] \right]$$

$$= \sum_{t=0}^{\infty} \sum_{i=1}^{n^t} \mathbb{E}_{s_{t,i},(a_t)_{1:n}^{(i)}} \left[ Q_{\pi_\beta}^\sharp (s_t^{(i)}, (a_t)_{1:n}^{(i)}) \right]$$

$$= \mathbb{E}_{\pi_\theta} \left[ \sum_{t=0}^{\infty} \gamma^t \sum_{i=1}^{n^t} Q_{\pi_\beta}^\sharp (s_t^{(i)}, (a_t)_{1:n}^{(i)}) \right]$$

Therefore,

$$V_{\pi_\theta}^\sharp (s) - V_{\pi_\beta}^\sharp (s)$$

$$= \mathbb{E}_{\pi_\theta} \left[ \sum_{t=0}^{\infty} \gamma^t \sum_{i=1}^{n^t} Q_{\pi_\beta}^\sharp (s_t^{(i)}, (a_t)_{1:n}^{(i)}) - V_{\pi_\beta}^\sharp (s_t^{(i)}) \,\Big|\, s_0 = s \right].$$

Then, using eq. (8), the statement follows.

$\square$

## D.2  PROOF OF PROPOSITION 5.1

We follow the set-up in Cui et al. (2025). We parameterize the policy as a softmax distribution:

$$\pi_\theta(a \mid s) = \frac{\exp(z_{sa})}{\sum_{a'} \exp(z_{sa'})}$$

where $z_{sa}$ is the output logit of action $a$ from state $s$. We also have the following derivative:

$$\frac{\partial}{\partial z_{sa'}} \log \pi_\theta(a \mid s) = 1\{a' = a\} - \pi_\theta(a' \mid s).$$

First, we prove the following that shows how the output logit changes after one-step parameter update in the set RL paradigm.

**Lemma D.2** *Given the softmax policy $\pi_\theta$ is updated using eq. (3) using a step-size $\alpha$, the change in the output logit $z_{sa}$ after one step update is given by*

$$z_{sa}^{k+1} - z_{sa}^k = \alpha \mathbb{E}_{a_{1:n} \sim \pi_\theta^k(\cdot|s)} \left[ f(s, a_{1:n}) \sum_{i=1}^n 1\{a_i = a\} \right] - \alpha n \pi_\theta^k(a \mid s) \mathbb{E}_{a_{1:n} \sim \pi_\theta^k(\cdot|s)} \left[ f(s, a_{1:n}) \right].$$

*Proof.* This can be proven by elementary properties of expectation:

$$\begin{aligned}
z_{sa}^{k+1} - z_{sa}^k &= \alpha \frac{\partial}{\partial z_{sa}^k} \mathbb{E}_{a_{1:n} \sim \pi_\theta^k(\cdot|s)} \left[ f(s, a_{1:n}) \right] \\
&= \alpha \mathbb{E}_{a_{1:n} \sim \pi_\theta^k(\cdot|s)} \left[ \frac{\partial}{\partial z_{sa}^k} \log(\mathbb{P}(a_{1:n} \mid s)) f(s, a_{1:n}) \right] \\
&= \alpha \mathbb{E}_{a_{1:n} \sim \pi_\theta^k(\cdot|s)} \left[ \sum_{i=1}^n \frac{\partial}{\partial z_{sa}^k} \log(\pi_\theta^k(a_i \mid s)) f(s, a_{1:n}) \right] \\
&= \alpha \mathbb{E}_{a_{1:n} \sim \pi_\theta^k(\cdot|s)} \left[ \sum_{i=1}^n \left( 1\{a_i = a\} - \pi_\theta^k(a \mid s) \right) f(s, a_{1:n}) \right]
\end{aligned}$$

$\square$

Now we prove the main result:

**Proposition 5.1 (restated).** Consider the set-RL setup at state $s$. After one update to the policy, the change in entropy, $\Delta = \mathcal{H}\left(\pi_\theta^{k+1} \mid s\right) - \mathcal{H}\left(\pi_\theta^k \mid s\right)$, is given by

$$\Delta \approx -\alpha \mathrm{Cov}_{a_{1:n}} \left( \frac{1}{n} \sum_{i=1}^n \log \pi_\theta^k(a_i \mid s), \mathrm{Cov}_{a_{1:n}'} \left( f(s, a_{1:n}'), \sum_{i,j=1}^n 1\{a_i = a_j'\} \right) \right),$$

where both covariances are taken with respect to $\pi_\theta^k(\cdot \mid s)$.

*Proof.* We have the following first order approximation of $\Delta$ (Cui et al., 2025):

$$\Delta \approx -\mathrm{Cov}_{a \sim \pi_\theta^k(\cdot|s)} \left( \log \pi_\theta^k(a \mid s), z_{sa}^{k+1} - z_{sa}^k \right).$$

Using Lemma D.2, we get that

$$\begin{aligned}
\Delta \approx & \alpha n \mathbb{E}[f(s, a_{1:n})] \mathrm{Cov}_{a \sim \pi_\theta^k(\cdot|s)} \left( \log \pi_\theta^k(a \mid s), \pi_\theta^k(a \mid s) \right) \\
& - \alpha \mathrm{Cov}_{a \sim \pi_\theta^k(\cdot|s)} \left( \log \pi_\theta^k(a \mid s), \mathbb{E}_{a_{1:n} \sim \pi_\theta^k(\cdot|s)} \left[ f(s, a_{1:n}) \sum_{i=1}^n 1\{a_i = a\} \right] \right)
\end{aligned}$$

Note that

$$\begin{aligned}
\mathrm{Cov}_{a_{1:n}} \left( f(s, a_{1:n}), \sum_{i=1}^n 1\{a_i = a\} \right) = & \mathbb{E}_{a_{1:n} \sim \pi_\theta^k(\cdot|s)} \left[ f(s, a_{1:n}) \sum_{i=1}^n 1\{a_i = a\} \right] \\
& - n \pi_\theta^k(a \mid s) \mathbb{E}_{a_{1:n} \sim \pi_\theta^k(\cdot|s)} \left[ f(s, a_{1:n}) \right]
\end{aligned}$$

Using this, we get that

$$\begin{aligned}
\Delta \approx & \alpha n \mathbb{E}[f(s, a_{1:n})] \mathrm{Cov}_{a \sim \pi_\theta^k(\cdot|s)} \left( \log \pi_\theta^k(a \mid s), \pi_\theta^k(a \mid s) \right) \\
& - \alpha \mathrm{Cov}_{a \sim \pi_\theta^k(\cdot|s)} \left( \log \pi_\theta^k(a \mid s), \mathbb{E}_{a_{1:n} \sim \pi_\theta^k(\cdot|s)} \left[ f(s, a_{1:n}) \sum_{i=1}^n 1\{a_i = a\} \right] \right) \\
= & \alpha n \mathbb{E}[f(s, a_{1:n})] \mathrm{Cov}_{a \sim \pi_\theta^k(\cdot|s)} \left( \log \pi_\theta^k(a \mid s), \pi_\theta^k(a \mid s) \right)
\end{aligned}$$

$$- \alpha \mathrm{Cov}_{a \sim \pi_\theta^k(\cdot|s)} \left( \log \pi_\theta^k(a \mid s), \mathrm{Cov}_{a_{1:n}} \left( f(s, a_{1:n}), \sum_{i=1}^n 1\{a_i = a\} \right) + \right.$$

$$\left. n \pi_\theta^k(a \mid s) \mathbb{E}_{a_{1:n} \sim \pi_\theta^k(\cdot|s)} [f(s, a_{1:n})] \right)$$

$$= -\alpha \mathrm{Cov}_{a \sim \pi_\theta^k(\cdot|s)} \left( \log \pi_\theta^k(a \mid s), \mathrm{Cov}_{a_{1:n}} \left( f(s, a_{1:n}), \sum_{i=1}^n 1\{a_i = a\} \right) \right).$$

All that remains to show is

$$\mathrm{Cov}_{a_{1:n}} \left( \frac{1}{n} \sum_{i=1}^n \log \pi_\theta^k(a_i \mid s), \mathrm{Cov}_{a'_{1:n}} \left( f(s, a'_{1:n}), \sum_{i,j=1}^n 1\{a'_i = a_j\} \right) \right)$$

$$= \mathrm{Cov}_{a \sim \pi_\theta^k(\cdot|s)} \left( \log \pi_\theta^k(a \mid s), \mathrm{Cov}_{a_{1:n}} \left( f(s, a_{1:n}), \sum_{i=1}^n 1\{a_i = a\} \right) \right).$$

This can be seen using the linearity of covariance and the fact that each $a_i$ in $a_{1:n}$ is sampled independently:

$$\mathrm{Cov}_{a_{1:n}} \left( \frac{1}{n} \sum_{i=1}^n \log \pi_\theta^k(a_i \mid s), \mathrm{Cov}_{a'_{1:n}} \left( f(s, a'_{1:n}), \sum_{i,j=1}^n 1\{a_i = a'_j\} \right) \right)$$

$$= \frac{1}{n} \sum_{i=1}^n \mathrm{Cov}_{a_{1:n}} \left( \log \pi_\theta^k(a_i \mid s), \mathrm{Cov}_{a'_{1:n}} \left( f(s, a'_{1:n}), \sum_{j=1}^n 1\{a_i = a'_j\} \right) \right)$$

$$= \mathrm{Cov}_{a \sim \pi_\theta^k(\cdot|s)} \left( \log \pi_\theta^k(a \mid s), \mathrm{Cov}_{a_{1:n}} \left( f(s, a_{1:n}), \sum_{i=1}^n 1\{a_i = a\} \right) \right).$$

$\square$

### D.3 PROOF OF LEMMA 5.3

**Lemma 5.3 (restated).** Consider any set of actions $a_{1:n} \in \mathcal{A}^n$. We can write the change in the log probability of sampling this set of actions after one policy update using set RL as:

$$\log \pi_\theta^{k+1}(a_{1:n} \mid s) = \log \pi_\theta^k(a_{1n} \mid s) + \alpha \Lambda_f(a_{1:n}; \pi_\theta^k) - \alpha C(\theta^k)$$

where $C(\theta^k)$ is a function independent of $a_{1:n}$.

*Proof.* This follows from using a first-order Taylor approximation of $\sum_{i=1}^n \log \pi_\theta^{k+1}(a_i \mid s)$ at $\sum_{i=1}^n \log \pi_\theta^k(a_i \mid s)$ and then using Lemma D.2. Here, $C(\theta^k) = \mathrm{Cov}_{a'_{1:n} \sim \pi_\theta^k(\cdot|s)} \left( f(s, a'_{1:n}), \sum_{i=1}^n \pi_\theta^k(a'_{1:n} \mid s) \right)$. $\square$

### D.4 PROOF OF PROPOSITION 5.4

**Proposition 5.4 (restated).** Consider the polychromic objective in eq. (6). For any homogeneous set $a_{1:n} = \{a\}$ where $r(s, a) = 1$, there exists $\epsilon \in (0, 1)$ such that $\Lambda_{f_{\mathrm{poly}}}(a) < 0$ when $\pi_\theta(a \mid s) > \epsilon$. Furthermore, the scaffold values of these homogeneous sets satisfy the bound $\Lambda_{f_{\mathrm{poly}}}(a) \leq \sqrt{\frac{p(1-p)}{n}}$.

*Proof.* We first prove the first part of the proposition. Let $p = \pi_\theta(a \mid s)$. We will use the shorthand $\Lambda(a) = \Lambda_{f_{\mathrm{poly}}}(a; \pi_\theta)$, $f = f_{\mathrm{poly}}$ and $\hat{f} = \mathbb{E}_{a_{1:n} \sim \pi_\theta(\cdot|s)}[f(s, a_{1:n})]$. Then,

$$\Lambda(a) = \mathrm{Cov}_{a'_{1:n} \sim \pi_\theta(\cdot|s)}(\hat{f}(s, a'_{1:n}), \frac{1}{I(a)} \sum_{i,j=1}^n 1\{a'_i = a_j\})$$

$$= \mathrm{Cov}_{a'_{1:n} \sim \pi_\theta(\cdot|s)}(\hat{f}(s, a'_{1:n}), \frac{1}{n} \sum_{i=1}^n 1\{a'_i = a\})$$

$$= \sum_{j=0}^{n} \left( \sum_{|a'_{1:n} \cap \{a\}|=j} \pi_\theta(a'_{1:n} \mid s)(f(s, a'_{1:n}) - \hat{f})(\frac{j}{n} - \mathbb{E}_{\alpha_{1:n} \sim \pi_\theta(\cdot|s)} \left[ \frac{1}{n} \sum_{i=1}^{n} 1\{\alpha_i = a\} \right]) \right)$$

$$= \sum_{j=0}^{n} \left( \sum_{|a'_{1:n} \cap \{a\}|=j} \pi_\theta(a'_{1:n} \mid s)(f(s, a'_{1:n}) - \hat{f})(\frac{j}{n} - p) \right)$$

$$= \sum_{j=0}^{\lfloor np \rfloor} (\frac{j}{n} - p) \left( \sum_{|a'_{1:n} \cap \{a\}|=j} \pi_\theta(a'_{1:n} \mid s)(f(s, a'_{1:n}) - \hat{f}) \right)$$

$$+ \sum_{j=\lfloor np \rfloor+1}^{n} (\frac{j}{n} - p) \left( \sum_{|a'_{1:n} \cap \{a\}|=j} \pi_\theta(a'_{1:n} \mid s)(f(s, a'_{1:n}) - \hat{f}) \right)$$

$$= \sum_{j=0}^{\lfloor np \rfloor} (\frac{j}{n} - p) \left( \sum_{|a'_{1:n} \cap \{a\}|=j} \pi_\theta(a'_{1:n} \mid s)(f(s, a'_{1:n})) \right)$$

$$+ \sum_{j=\lfloor np \rfloor+1}^{n} (\frac{j}{n} - p) \left( \sum_{|a'_{1:n} \cap \{a\}|=j} \pi_\theta(a'_{1:n} \mid s)(f(s, a'_{1:n})) \right)$$

$$- \hat{f} \left( \sum_{j=0}^{n} \sum_{|a'_{1:n} \cap \{a\}|=j} \pi_\theta(a'_{1:n} \mid s)(\frac{j}{n} - p) \right)$$

Now, we can simplify using properties of the binomial distribution as follows:

$$\sum_{j=0}^{n} \sum_{|a'_{1:n} \cap \{a\}|=j} \pi_\theta(a'_{1:n} \mid s)(\frac{j}{n} - p) = \sum_{j=0}^{n} \left( \frac{j}{n} - p \right) \binom{n}{j} p^j (1-p)^{n-j} = 0.$$

Therefore, the scaffold value becomes:

$$\Lambda(a) = \sum_{j=0}^{\lfloor np \rfloor} (\frac{j}{n} - p) \left( \sum_{|a'_{1:n} \cap \{a\}|=j} \pi_\theta(a'_{1:n} \mid s)(f(s, a'_{1:n})) \right)$$

$$+ \sum_{j=\lfloor np \rfloor+1}^{n} (\frac{j}{n} - p) \left( \sum_{|a'_{1:n} \cap \{a\}|=j} \pi_\theta(a'_{1:n} \mid s)(f(s, a'_{1:n})) \right)$$

$$\leq \sum_{j=0}^{\lfloor np \rfloor} (\frac{j}{n} - p) \frac{2j}{n^2} \sum_{|a'_{1:n} \cap \{a\}|=j} \pi_\theta(a'_{1:n} \mid s)$$

$$+ \sum_{j=\lfloor np \rfloor+1}^{n} (\frac{j}{n} - p) \frac{n-j+1}{n} \sum_{|a'_{1:n} \cap \{a\}|=j} \pi_\theta(a'_{1:n} \mid s)$$

$$= \sum_{j=0}^{\lfloor np \rfloor} (\frac{j}{n} - p) \frac{2j}{n^2} \binom{n}{j} p^j (1-p)^{n-j}$$

$$+ \sum_{j=\lfloor np \rfloor+1}^{n} (\frac{j}{n} - p) \frac{n-j+1}{n} \binom{n}{j} p^j (1-p)^{n-j}$$

$$= \sum_{j=0}^{\lfloor np \rfloor} (\frac{j}{n} - p) \frac{2j}{n^2} \binom{n}{j} p^j (1-p)^{n-j}$$

$$+ \sum_{j=\lfloor np \rfloor+1}^{n-1} (\frac{j}{n} - p) \frac{n-j+1}{n} \binom{n}{j} p^j (1-p)^{n-j}$$

Here, in the second line, we used the fact when $a'_{1:n} \cap \{a\}$ is of size $j$, then the smallest possible value of $f(s, a'_{1:n})$ is $\frac{2j}{n^2}$ since at least 2 out of $n$ elements are unique and at least $j$ out of $n$ elements attain reward $+1$. On the other hand, we can bound it above by $\frac{n-j+1}{n}$ which happens if all elements get reward $+1$ and $n - j + 1$ elements are unique. In the last line, we used the fact that when $f(s, a'_{1:n} = \{a\}) = 0$ as the diversity is 0. Now, let $\epsilon = \frac{n-1}{n}$. Then, when $p \geq \epsilon$, $\lfloor np \rfloor \geq n - 1$. Therefore,

$$\Lambda(a) = \sum_{j=0}^{\lfloor np \rfloor} (\frac{j}{n} - p) \frac{2j}{n^2} \binom{n}{j} p^j (1-p)^{n-j} \leq 0.$$

Now we prove the second part. First, define the following upper bound of the scaffold value of the homogeneous sets:

$$\Lambda(a) \leq \mathbb{E}_{X \sim \text{Bin}(n,p)} \left[ \left( \frac{X}{n} - p \right) C_n(X) \right] =: B_a(n)$$

where $C_n(x) = \begin{cases} \frac{2x}{n^2} & x \leq \lfloor np \rfloor \\ \frac{n-x+1}{n} & x \in [\lfloor np \rfloor + 1, n-1] \\ 0, & x = n \end{cases}$ .

Now, $\mathbb{E}[\frac{X}{n} - p] = 0$ and $\mathbb{E}[(\frac{X}{n} - p)^2] = \text{Var}(\frac{X}{n}) = \frac{1}{n^2} \text{Var}(X) = \frac{p(1-p)}{n}$. On the other hand, we can bound $C_n(X)$ as follows: when $x \leq \lfloor np \rfloor$, $C_n(x) = \frac{2x}{n^2} \leq \frac{2\lfloor np \rfloor}{n^2} \leq \frac{2}{n}$. On the other hand, when $j \in [\lfloor np \rfloor + 1, n-1]$, we have $C_n(x) = \frac{n-x+1}{n} \leq 1$. Combining, we have that $\mathbb{E}[C_n(X)^2] \leq 1$. Therefore, using the Cauchy–Schwarz inequality:

$$\begin{aligned} B_a(n) &= \mathbb{E}_{X \sim \text{Bin}(n,p)} \left[ \left( \frac{X}{n} - p \right) C_n(X) \right] \\ &\leq \sqrt{\mathbb{E}\left[ \left( \frac{X}{n} - p \right)^2 \right] \mathbb{E}[C_n(X)^2]} \\ &\leq \sqrt{\frac{p(1-p)}{n}}. \end{aligned}$$

$\square$

### D.5   Proof of Proposition 5.5

**Proposition 5.5 (restated).** Suppose $a_{1:n}$ is heterogeneous where each $a_i$ is unique with probability $\frac{p}{n}$ where $p \in (0, \frac{1}{n})$. Suppose exactly $q$ of the $n$ actions satisfy $r(s, a_i) = 1$, and that any other action $a' \notin a_{1:n}$ with $\pi_\theta(a' \mid s) > 0$ yields $r(s, a') = 0$. Then, the scaffold value of $a_{1:n}$ satisfies $\Lambda_{f_{\text{poly}}}(a_{1:n}) > \frac{qp^n(1-p)}{n} \left(1 - \left(1 - \frac{1}{n}\right)^n\right)$.

*Proof.* This can be proven using very similar techniques. Let $P_j = \binom{n}{p} p^j (1-p)^{n-j}$. Ag ain, we use the shorthand $f = f_{\text{poly}}$. Then,

$$\begin{aligned} \Lambda(a_{1:n}) &= \sum_{j=0}^{n} P_j (\frac{j}{n} - p) \mathbb{E}_{|a'_{1:n} \cap a_{1:n}|=j} \left[ f(s, a'_{1:n}) \right] \\ &= P_0(0 - p) \cdot 0 + \sum_{j=1}^{\lfloor np \rfloor} P_j (\frac{j}{n} - p) \mathbb{E}_{|a'_{1:n} \cap a_{1:n}|=j} \left[ f(s, a'_{1:n}) \right] \\ &\quad + \sum_{j=\lfloor np \rfloor + 1}^{n} P_j (\frac{j}{n} - p) \mathbb{E}_{|a'_{1:n} \cap a_{1:n}|=j} \left[ f(s, a'_{1:n}) \right] \end{aligned}$$

$$\geq P_n(\frac{n}{n} - p)\mathbb{E}_{|a'_{1:n} \cap a_{1:n}|=n}\left[f(s, a'_{1:n})\right]$$

$$= P_n(1 - p)\mathbb{E}_{|a'_{1:n} \cap a_{1:n}|=n}\left[f(s, a'_{1:n})\right]$$

All that remains to show is that

$$\mathbb{E}_{|a'_{1:n} \cap a_{1:n}|=n}\left[f(s, a'_{1:n})\right] = \frac{q}{n}\left(1 - \left(1 - \frac{1}{n}\right)^n\right).$$

Note that, since the intersection's cardinality is $n$ and the sampling is done independently, this is equivalent to drawing $n$ items uniformly with replacement from $a_{1:n}$. Also note that $\mathbb{E}\left[\frac{1}{n}\sum_i r(s, a'_i) \cdot d(s, a'_{1:n})\right] = \frac{1}{n}\sum_i \mathbb{E}[r(s, a'_i) \cdot d(s, a'_{1:n})]$. Since this is a uniform distribution, this becomes equal to $\mathbb{E}[r(s, a'_i)]\mathbb{E}[d(s, a'_{1:n})] = \frac{q}{n}\mathbb{E}[d(s, a'_{1:n})]$.

On the other hand, to compute $\mathbb{E}[d(s, a'_{1:n})]$, we use indicator variables. Let $1_k = 1\{a_k \in a'_{1:n}\}$. Then, $\mathbb{E}\left[1_k\right] = 1 - \left(1 - \frac{1}{n}\right)^n$. Now,

$$\mathbb{E}_{|a'_{1:n} \cap a_{1:n}|=n}[d(s, a'_{1:n})] = \mathbb{E}_{|a'_{1:n} \cap a_{1:n}|=n}[\frac{1}{n}\sum_k 1_k]$$

$$= 1 - \left(1 - \frac{1}{n}\right)^n$$

$\square$

