# OpenReview forum: "Polychromic Objectives for Reinforcement Learning"
_ICLR.cc/2026/Conference — ICLR 2026 Poster_

### Official Review · Reviewer_Bmic · 2025-10-28

**Soundness:** 2
**Presentation:** 1
**Contribution:** 2
**Rating:** 2
**Confidence:** 3

**Summary:**

The work introduces the notion of set RL, in which agents are maximizing rewards not with respect to individual trajectories, but with respect to a set of trajectories. This formulation is appealing as it naturally allows to directly optimize for reward maximization as well as diversity. Based of the notion of set RL, the authors propose polychromic objectives, which give a practical way of opimizing both for success as well as diversity of the set of trajectories. The work then further discusses how to adapt PPO to work with this particular polychromic objective before evaluating it on three different environments. The Poly-PPO implementation seems ot generally improve over pretrained policies and outperforming baselines. Finally, the work provides a more theoretical analysis on the effect of polychromic objectives on the entropy of a policy.

**Strengths:**

The notion of set RL seems appealing and could inspire novel learning approaches that are distinct from existing classical RL algorithms. Further, the polychromic objective seems like a good logical consequence of set RL.

**Weaknesses:**

The work discusses various aspects fairly shallowly. In the beginning, the work reads like it is written just to introduce the idea of set RL. The work raises the expectation in the reader that a more comprehensive discussion of set RL and how it might differentiate itself from multi-objective RL. However, before such a discussion starts, the work pivots to discuss a particular aspect of set RL in the form of polychromic objectives. Before going into depth on polychromic objectives, a particular instantiation of PPO is discussed that makes use of this objective. While the work states in line 166 that the "generality of set RL" has been discussed, the overall work only discusses a very particular instantiation that seems to require a lot of adaptation from classical RL to work.
Overall, a lot of design decisions seem to just fall out of the blue without being adequately discussed. I fail to see why, e.g., no other RL algorithm is being considered for the extension to set RL. Similarly, why is there no ablation on the choice of diversity function for the Poly-PPO implementation? The choice of REINFORCE as a baseline is never justified, nor is the choice of environments.

Taken altogether, the work does not seem fit for publication in its current state and there are too many loose ends that need to be taken care of before I would consider increasing my score. I am happy to adjust my score if the authors show that I have misunderstood crucial aspects but vote for rejection as is.

**Questions:**

* How is set RL distinct from multi-objective RL? Specifically, how is the notion of "sets" different from pareto-fronts?
* Why did you focus only on PPO and no other RL algorithm?
* Is the choice of diversity function a crucial one or is Poly-PPO fairly insensitive to this choice?
* How did you set the hyperparameters of your method? Did you tune the baselines and your new algorithm?

---

> ### Author Response · Authors · 2025-11-18
> **Official Comment by Authors (1/2)**
>
> Thank you for your thoughtful feedback on what could be improved, especially on presentation. We respond to your questions below:
>
> ---
> > How is set RL distinct from multi-objective RL?
> - **They are defined over fundamentally different MDPs.** Our definition of set RL (much like standard/classical RL) is defined using a scalar valued reward function $r : \mathcal{S} \times \mathcal{A} \rightarrow \mathbb{R}$. In contrast, multi-objective RL (MORL, in short) is defined using a vector valued reward function $r : \mathcal{S} \times \mathcal{A} \rightarrow \mathbb{R}^d$ where $d$ is the number of objectives. This makes these two frameworks orthogonal to each other.
> - **They have different problem formulations.** In set RL, we aim to train a policy towards generating the optimal set of trajectories with respect to one objective function i.e.  $ \arg\max_\theta \mathbb{E}_{\tau_1,\cdots,\tau_n}\left[ f(s_0,\tau_1,\cdots, \tau_n) \right].$ Set RL uses a single objective defined over multiple trajectories.
> - In contrast, muti-objective RL aims to train a policy towards generating an optimal trajectory with respect to multiple objectives:  $ \arg\max_\theta \mathbb{E}_{\tau}\left[ \sum_t \gamma^t \vec{r}(s_t, a_t) \right],$ where $\vec{r}(s, a)$ is now vector-valued. Therefore, MORL uses multiple objectives, all of which are defined over the same, single trajectory.
> - Intuitively, set RL is explicitly optimizing for the generation of optimal set of trajectories, which MORL does not.
> - Finally, note that it is possible to optimize multiple objectives using set RL as well by using vector-valued objectives of the form $f(s, \tau_1,\cdots,\tau_n) \in \mathbb{R}^d$. In this case, there is still a crucial difference from MORL - set RL would still be using objectives defined over multiple trajectories whereas MORL’s objective (returns) is defined over a single.
>
> **We have added a discussion on these differences between set RL and multi-objective RL in section 3.1 (highlighted in blue).**
>
> ---
>
> > Why did you focus only on PPO and no other RL algorithm?
> - We interpret this question as asking why poly-PPO is built on PPO specifically. Note that in our experiments we do compare against other baselines, including REINFORCE and UCB variants.
> - PPO is widely used in RL fine-tuning settings - including RL for reasoning [1] and RLHF [2] - because it is stable and scalable in practice.
> - PPO also supports learning from mildly off-policy data via importance sampling and clipping, which improves sample efficiency and training stability [3].
> - Finally, the performance difference lemma motivates PPO in standard RL [4]. Since we extend this lemma to the set-RL setting (Lemma 3.2), it is natural to develop a PPO-style algorithm for optimizing the polychromic objective.
>
> **We have added a discussion on why we modify PPO to optimize the polychromic objective in Section 3.3.**
>
> ---
>
> > How did you set the hyperparameters of your method? Did you tune the baselines and your new algorithm?
> - We report the hyperparameters of poly-PPO in Table 3. We tuned the hyperparameters for PPO, REINFORCE, their UCB variants as well as our method. We tuned all hyperparameters by running each configuration across 5 seeds. Note that the final performance reported in the paper for all experiments is with the fixed, final set of hyperparameters.
>
> ---
>
> > Is the choice of diversity function a crucial one or is Poly-PPO fairly insensitive to this choice?
> - The diversity function specifies what kind of diversity the policy is encouraged to produce. For instance, if it only measures local action-level variation, Poly-PPO will not promote trajectory-level exploration; if it measures only outcome-level diversity, the policy will seek diverse outcomes. Thus, the appropriate choice depends on the domain.
> - Our paper does not aim to solve the problem of designing optimal diversity functions. Prior work has proposed many such measures, e.g., Vendi Score [7], for evaluation [5], uncertainty analysis [8], RL [6] etc. Our contribution is complementary: given a diversity function, we study how to use it to optimize for sets of trajectories that are both diverse and successful. This is analogous to papers on standard RL algorithms (e.g. PPO), which assume access to a reward function rather than jointly tackling reward design and policy optimization.
>
> ---
> References:
> 1. Math-Shepherd. arXiv:2312.08935
> 2. Training language models to follow instructions with human feedback. arXiv:2203.02155
> 3. Proximal Policy Optimization. https://spinningup.openai.com/en/latest/algorithms/ppo.html
> 4. Trust region policy optimization. arXiv:1502.05477
> 5. NoveltyBench. arXiv:2504.05228
> 6. Jointly Reinforcing Diversity and Quality. arXiv:2509.02534v1
> 7. Vendi Score. arXiv:2210.02410
> 8. Semantically diverse language generation. arXiv:2406.04306

---

> ### Author Response · Authors · 2025-11-18
> **Official Comment by Authors (2/2)**
>
> In addition to our earlier comment, we would also like to address some other points raised, focusing on (i) the aspects of set RL that we analyze in depth throughout the paper, and (ii) our choice of baselines.
>
> ---
> > The work raises the expectation in the reader that a more comprehensive discussion of set RL
> 1. We demonstrated how set RL can be optimized using policy gradient estimators with an unbiased baseline that does not discriminate trajectories within a set. We discuss how this framework is different from $n$-sample RL (Tang et. al.) whose gradient estimators discriminate between trajectories within a set.
> 2. We strengthen the understanding of what set RL aims to accomplish by proving the performance difference lemma in the set RL paradigm which helps us characterize policy improvement.
> 3. We theoretically analyse the change in entropy of a policy being trained using set RL (Proposition 5.1) and show that there is an “implicit individualized credit assignment” that comes out of set RL despite optimizing over sets (see paragraph after proposition 5.1).
>
> > Before going into depth on polychromic objectives, a particular instantiation of PPO is discussed that makes use of this objective.
> 1. After defining the objective in equation 6, we remarked that, given it is a set objective, it enables learning to explore trajectories that are not yet successful due to shared advantage over a set.
> 2. We later return to a deeper theoretical analysis. We analyze how the objective prevents entropy collapse onto a homogeneous set of behaviors (Proposition 5.3) and encourages the upweighting of heterogeneous sets (Proposition 5.4).
>
> ---
> > The choice of REINFORCE as a baseline is never justified, nor is the choice of environments
> 1. In this paper, we investigated a few different baselines: REINFORCE, REINFORCE with UCB bonus, PPO, and PPO with UCB bonus. The choice of REINFORCE is natural as it is a standard and canonical algorithm; simple modifications of it are used in large-scale systems with remarkable success [1, 2]. Furthermore, REINFORCE is a principled approach to policy gradient algorithms since it directly optimizes the policy gradient. Therefore, we believe not comparing to REINFORCE would be unnatural. Lastly, it is quite clear from the empirical results that REINFORCE is a strong baseline, often outperforming PPO especially in BabyAI tasks.
> 2. Also note that our paper is concerned with on-policy reinforcement learning methods where exploration is a significantly greater challenge, not off-policy q-learning style methods. As such, it is not clear what alternative RL algorithm we should be incorporating as a baseline.
> 3. One popular algorithm we did leave out is GRPO. First of all, GRPO has several strong similarities to PPO except it aims to treat the problem as a bandit problem (even though it is not) by assigning the same advantage to all tokens/actions along a trajectory and does not train a learned critic. In both MiniGrid and BabyAI, it does not make sense at all to say all actions in a trajectory deserve the same credit (i.e. advantage). Furthermore, we are already comparing PPO and REINFORCE.
> 4. We evaluated all methods on BabyAI, Minigrid and Algorithmic Creativity. These environments evaluate agents on tasks that (a) require exploration to discover solutions, (b) allow for a diverse set of solutions and (c) allow for resets, which makes them strong testbeds for our specific topic.
>
> ---
> References:
> 1. MiniMax-M1.arXiv:2506.13585
> 2. Art of Scaling Reinforcement Learning. arXiv:2510.13786

---

> ### Comment · Reviewer_Bmic · 2025-11-20
>
> Thank you very much for the detailed response.
> After having read all reviews and the available rebuttals, I remain unconvinced and won't increase my score.
>
> The discussion on the difference of "Set RL" and multi-objective RL seems wrong to me. For example, it is said that MO-RL requires a vector of rewards. That's simply not true. Scalarization is commonly done in many multi-objective settings. I would strongly argue that the proposed Set RL is essentially using a form scalarization to jointly optimize the reward signal and diversity of trajectories.
>
> The response on the choice of diversity function also seems to not be aligned with MO-RL and Set-RL being different. If evaluating/proposing different diversity functions is seen as equivalent to reward design then I would see Set-RL simply as a particular instantiation of of MO-RL.
> What would stop a MO-RL approach from using multiple rollouts of the same policy to maximize a second diversity objective?

---

> ### Author Response · Authors · 2025-11-20
> **Official Comment by Authors**
>
> Thank you for responding. There is a misunderstanding here regarding what set RL is. **Set RL is not, by definition, equivalent to multi-objective RL**.
>
> ---
> **Difference 1**: **set RL need not be defined with respect to multiple objectives where multi-objective RL, by definition, is**.
> - As pointed out in section 3.1, set RL can optimize a *single objective* too -- there are other objectives, apart from the polychromic objective, that set RL can use. For example, $f(s_0, \tau_1,\cdots, \tau_n) = \frac{1}{n}\sum_{i=1}^n R(\tau_i)$, as pointed out in line 116-118, is simply a *single objective defined over a set of trajectories*.
> - Furthermore, one can optimize $n$-sample objectives [1] for inference-time performance such as (a) pass@$k$ performance with the objective $f(s_0, \tau_1,\cdots,\tau_n) = \max \left( R(\tau_1), \cdots, R(\tau_n) \right)$ and (b) majority-voting performance with the objective $f(s_0, \tau_1,\cdots,\tau_n) = R \left( \text{maj}(\tau_1,\cdots,\tau_n) \right)$.
>
> **Difference 2**: **set RL is defined over multiple trajectories whereas multi-objective RL is not**
> - Set Reinforcement Learning formulation: $\arg\max_\theta \mathbb{E}_{\tau_1,\cdots,\tau_n }[f(s_0, \tau_1,\cdots, \tau_n)]$.
> - Multi-objective reinforcement learning: $\arg\max_\theta \mathbb{E}_{\tau}[f(s_0, \tau)]$.
> - The distinguishing feature is that in set RL, an objective is defined over a *set* of trajectories. As we pointed out in section 3.1 and in our previous comments, we are optimizing for the optimal set of trajectories. Regardless of whether we scalarize the vector-valued objective or not, in multi-objective RL, the objective is defined over a single trajectory. As such, we would strongly claim that set RL is not just a scalarization of traditional multi-objective RL.
>
> **Difference 3**: **set RL provides a shared credit assignment to all trajectories within a set. Both standard RL and multi-objective RL provide individualized credit assignment to each trajectory**.
> - As pointed out in section 3.1’s discussion on the baseline, a trajectory receives a credit assignment because of its appearance in the set that it belongs to. For example, if we consider the objective $f(s_0, \tau_1,\cdots, \tau_n) = \frac{1}{n}\sum_{i=1}^n R(\tau_i)$, even if the trajectory $\tau_1$ receives return $R(\tau_1) = 0$, it may receive a positive credit assignment simply because the remaining trajectories $\tau_1,\cdots,\tau_n$ all received return $1$.
> - This is a crucial distinguishing feature that sets set RL apart from other works [1] that consider objectives defined over multiple trajectories. It is certainly not a feature of multi-objective RL.
>
> **Difference 4**: **Set RL requires the ability to sample a set of trajectories from a state thereby requiring a fixed initial state and/or ability to reset the environment. Multi-objective RL does not**.
> - set RL *requires the ability to generate multiple trajectories from a state - which requires the ability to reset the environment at least to a fixed initial state and preferably more*. This is primarily the reason why we incorporated vine sampling in our algorithm. Multi-objective RL does not place such a constraint and does not necessitate either a fixed initial state or vine sampling.
>
> We would appreciate a reference that shows MORL is not different from Set RL in all these four ways.
>
> ---
> > “What would stop a MO-RL approach from using multiple rollouts of the same policy to maximize a second diversity objective”
> - We believe that each framework (RL, set RL,MORL) needs a clear definition with respect to fundamental, *necessary* features.
> - **Set RL *needs to be defined over a set of trajectories*. MORL does not. Set RL *needs to have shared credit assignment across a set of trajectories*. MORL does not. MORL *needs multiple objectives* but set RL does not.** These defining features give rise to different algorithms (e.g, set RL naturally demands vine-sampling, MORL does not)
> - **Set RL is differentiated from MO-RL for similar reasons as why standard RL is differentiated from MO-RL**. The fundamental, *necessary* features are different. For the same reason, we differentiate multi-objective RL from standard RL. In multi-objective RL, we have $d$ objectives to optimize, whereas in standard RL we have only one objective (return maximization). One can of course claim that standard RL is just a particular instantiation of multi-objective RL with $d =1$. However, clearly they give rise to algorithms that have different desiderata in these two different frameworks and, clearly, we distinguish these frameworks.
> - Similarly, set RL *requires* objectives over a *set of trajectories all of whom must share credit assignment* and *does not require multiple objectives*. So, it should be defined separately.
>
> Set RL is distinct but complementary to MORL; we can add an extended discussion on connections if you refer us to relevant papers.
>
> [1] Inference Time Objectives. arXiv:2503.19595

---

> > ### Comment · Reviewer_Bmic · 2025-11-24
> >
> > Thank you for the continued dialogue. I am not trying to argue against set RL as such but I understand how my previous comment has given the impression.
> >
> > My line of questioning for MORL is more to provide a better baseline to highlight that set RL provides a unique advantage that might not be filled with existing methods. I do understand that both differ in their definitions and how they try to solve their respective problems. However, MORL could be used to find a policy that simultaneously maximizes expected rewards and rollout diversity.

---

> ### Author Response · Authors · 2025-11-25
>
> Thank you for clarifying your questions.
>
> > My line of questioning for MORL is more to provide a better baseline to highlight that set RL provides a unique advantage that might not be filled with existing methods. I do understand that both differ in their definitions and how they try to solve their respective problems. However, MORL could be used to find a policy that simultaneously maximizes expected rewards and rollout diversity.
>
> - While we agree that MORL *could* be used to simultaneously encourage success and diversity, we are unsure of which baseline we should be including here that would provide a better point of comparison than PPO/REINFORCE/UCB. Can you please point us to a reference with a baseline you would like to see us incorporate in our paper?
> - Note that for all our current policy-gradient baselines (PPO and REINFORCE), we tuned the entropy bonus, which is, in some sense, an approximation of a MORL algorithm (scalarizing by summing the objectives). Similarly, the UCB variants use an additional UCB bonus (explained in line 284-285 of section 4) which can also be seen as a scalarization of a MORL framework via summing. For example, the success rate ablation over the UCB bonus on REINFORCE is as follows (PPO results are in Table 1, not including them here due to space constraint and because UCB bonus helps REINFORCE significantly more than it helps PPO):
> | Task      | REINFORCE (UCB coeff 0) | REINFORCE (UCB coeff 0.0005) | REINFORCE (UCB coeff 0.005) | Poly-PPO (no UCB) | Poly-PPO (UCB coeff 0.0005) |
> |-----------|---------------------------|-------------------------------|------------------------------|--------------------|------------------------------|
> | Bosslevel | 33.4 $\pm$ 1.4            | 36.4 $\pm$ 2.7                | 33.0 $\pm$ 4.2               | 45.2 $\pm$ 1.6     | 46.8 $\pm$ 1.6               |
> | Synthseq  | 45.4 $\pm$ 2.5            | 47.8 $\pm$ 2.4                | 42.9 $\pm$ 3.2               | 47.0 $\pm$ 3.1     | 43.2 $\pm$ 3.1               |
>
> - Both the UCB bonus and entropy bonus are used in order to encourage simultaneous maximization of returns and exploration, as explained in several works cited in our paper [1, 2, 3]
>
> [1] Reinforcement learning with deep energy-based policies. arXiv:1702.08165
> [2] Soft actor-critic. arXiv:1801.01290
> [3] Outcome-based Exploration for LLM Reasoning. arXiv:2509.06941

---

### Official Review · Reviewer_tLb8 · 2025-10-30

**Soundness:** 3
**Presentation:** 2
**Contribution:** 3
**Rating:** 6
**Confidence:** 3

**Summary:**

This paper studies how to induce diverse generations (trajectories) from a
policy trained with reinforcement learning (RL). The motivation is that current
policies trained with RL finetuning (RLFT) tend to collapse to a single mode of
behavior. One symptom of this is that increasing the number of trials afforded
to an RL policy does not increase its performance (i.e., pass@n coverage is
low), because similar trajectories are generated.

Thus, this paper proposes to create RL policies that generate diverse,
high-performing trajectories. This approach begins with defining set RL, an
extension of RL that considers objectives over a set of trajectories created by
a policy rather than over a single trajectory. Within set RL, the paper then
proposes a "polychromic objective" that motivates the trajectories generated by
a policy to be diverse and high-performing. Finally, the paper solves this
objective with an extension of PPO, referred to as "polychromic PPO."
Experiments are conducted on Minigrid, BabyAI, and Algorithmic Creativity,
showing that polychromic PPO can finetune policies to generate trajectories with
higher diversity, as evinced by higher pass@n scores.

**Strengths:**

1. I found the description of set RL and the polychromic objective intuitive and
   easy to follow. Since I am not as familiar with this field, I do not know how
   novel this objective is. However, I personally have not seen anything like it
   before. The closest I am aware of are works that seek to find diverse
   policies, like Diversity is All You Need and quality diversity algorithms.
   These works seek to find a set of diverse policies, while the current paper
   trains a single policy that can generate diverse trajectories.
1. I think the experimental evaluation is quite thorough, in that it seeks to
   answer a number of important questions about polychromic PPO, and it directly
   addresses the pass@n issue initially raised in the introduction.

**Weaknesses:**

1. The experiments are only run over 3 seeds, which is quite few. Furthermore,
   there do not seem to be error bars, and statistical testing was not performed
   to verify significant differences. Including these things would make the
   experiments more precise, e.g., statistical significance could be discussed
   instead of saying "achieves slightly lower validity" on lines 313-314. Below
   I list a couple of other minor issues with the presentation of the
   experiments:
   1. Table 1: It is unclear what values are bolded and why.
   1. Figure 2 could be cleaned up a bit. Namely:
      - The text is small and hard to read.
      - The y-axis bounds should bound the values in the graph, e.g., the graphs
        in the leftmost column should have bounds from 30% to 100% rather than
        30% to 90%.
      - The rows and columns should be labeled, instead of having the names only
        be in the caption.
   1. Figure 1 appears after Figure 2 in the paper.
   1. Line 307 is a comma splice.
   1. Figure 3 also has really small text.
1. The Entropy Analysis in Section 5 seems to provide valuable insights into how
   polychromic objectives work, but I found it difficult to follow, in part
   because I do not have background on what "entropy collapse" means. Perhaps
   some background could be provided on why we care to analyze the entropy of
   the policy?

**Questions:**

Below are minor comments and questions I found while reading the paper:

1. Please use the correct citation format for in-text citations. They should be
   author-year rather than numbers. Per the instructions in the ICLR template:

   > Citations within the text should be based on the natbib package and include
   > the authors' last names and year (with the et al. construct for more than
   > two authors).

1. In Eq. 2, is $s_0$ assumed to be fixed? $s_0$ is not really discussed when
   Eq. 2 is introduced.
1. Line 153-154: grammar mistake; maximize is used twice in the sentence?
1. I found it a bit confusing that the paper mentions Minigrid and BabyAI as two
   separate domains on which polychromic PPO is evaluated (line 64, 249). This
   is confusing because the results only seem to be shown for BabyAI (Table 1)
   and not Minigrid; furthermore, Section 4.2 only discusses BabyAI and not
   Minigrid. Are Minigrid and BabyAI the same benchmark?
1. Diversity in the experiments (Section 4.2) seems to be measured indirectly by looking at the pass@n ratings.
   Is it possible to measure diversity directly by sampling multiple rollouts of
   the policy and computing the diversity of those trajectories, e.g., using the
   diversity functions described on lines 263-264?

---

> ### Author Response · Authors · 2025-11-22
> **Official Comment by Authors 1**
>
> Thank you for your thoughtful feedback. We respond to your questions below:
>
> > Perhaps some background could be provided on why we care to analyze the entropy of the policy?
> - Entropy collapse refers to the phenomenon of a policy’s entropy decreasing as it keeps gathering the probability mass (of the policy distribution) on some subset of actions. Typically, in standard RL using policy gradient methods, prior work [1] demonstrate that policies collapse their entropy on to those actions that yield high returns i.e., they show that $\mathcal{H}(\pi_\theta^{k+1} \mid s) - \mathcal{H}(\pi_\theta^{k} \mid s) \approx \alpha \mathrm{Cov}(\log \pi_\theta^k(a \mid s), A(s, a)).$ The value of this analysis is in understanding exploration during RL – if the policy’s entropy is just almost monotonically decreasing, the policy is essentially not exploring to discover alternative methods of solving the task and, instead, collapsing on some behavior that already yields positive advantage. Our work aims to train a policy to explore and learn a diverse set of behaviors that are successful, and such exploration requires the policy to not monotonically collapse its entropy.
> - In our paper, we first analyze how the entropy changes when we optimize in set RL using any objective function $f(s, a_1,\cdots,a_n)$ (not necessarily the polychromic objective) with policy gradient methods and this yields Proposition 5.1.
> - This entropy analysis lends us a few helpful observations. For example, there is an implicit individualized credit assignment. For example, consider a completely homogeneous set i.e. the set $\{a_1,\cdots,a_n\} = \{a’\}$ where $a’$  is some successful action. Plug this action set into the polychromic objective, we can notice that since the diversity is 0, the polychromic objective’s value of this action set is 0. While one might suspect that in our set RL framework, since this action set receives a score 0, our policy never learns to upweight this successful action. However, this entropy analysis shows that the implicit credit assignment still leads to the policy learning to upweight this action at least until the loss in diversity overwhelms (Proposition 5.3).
> - More practically, this proposition suggests that, given we have sampled $B$ trajectories from a state, it is useful to construct sets combinatorially i.e. if we want sets of, say, $4$ trajectories in each, we might just combinatorially construct $\binom{B}{4}$ sets to get this implicit individual credit assignment (assuming scoring each set is not too expensive), whilst also being able to calculate an unbiased baseline.
>
> We have updated our paper with additional details motivating the entropy analysis (highlighted in blue).
>
> > The experiments are only run over 3 seeds, which is quite few.
> - We ran each method on each task over 3 seeds since every run takes about 20-25 hours.  Furthermore, for each method, the optimal hyperparameters were chosen over several seeds before we settled on the optimal set. The original PPO paper [2] also followed this set up (see section 6.1 from [2]). Furthermore, since these environments are all static environments and since, for all methods, we collected a large number of trajectories per data collection step, there is relatively less variance in performance for each method.
>
> > Are Minigrid and BabyAI the same benchmark?
> - They are not the same benchmark and this misunderstanding arose from our mistake in the caption for Table 1 which only mentioned BabyAI. We have updated the caption. Note that Goto, Pickup, Synthseq and Bosslevel are tasks in BabyAI whereas Four Rooms is a task from Minigrid.
>
> > Diversity in the experiments (Section 4.2) seems to be measured indirectly by looking at the pass@n ratings. Is it possible to measure diversity directly by sampling multiple rollouts of the policy and computing the diversity of those trajectories, e.g., using the diversity functions described on lines 263-264?
> - This could be a helpful way to visualize the diversity of the responses but it is agnostic to the performance of the policy. In our experiments, we found that the diversity is often maximal under policies that have struggled to discover solutions completely. For example, in the *Pickup* task, REINFORCE attains the largest diversity and in *Goto*, PPO attains the largest diversity. However, this is mainly because these methods struggled to learn successful solutions in several environments (as evident by their success rates in Table 1 and low pass@n coverage) and, as such, the agent keeps wandering around without making progress towards the goal. In contrast, the pass@$n$ curves are a more reliable way of measuring *effective diversity* i.e., the number of successful behaviors discovered and learned by the solution.
>
> References
>
> 1. Entropy mechanism of RL. arXiv:2505.22617
> 2. PPO. arXiv:1707.06347

---

> ### Author Response · Authors · 2025-11-22
> **Official Comment by Authors 2**
>
> In addition to our previous comment, we would like to thank the reviewer for providing us detailed suggestions on how to improve the quality of presentation of our paper. We have updated the pdf and changed the following as per the reviewer’s suggestions:
>
> 1. We changed the citation format to author-year format.
> 2. We cleaned up Figure 2 - we added error bars, updated the y-ticks that bound the y-axis, increased the font size of the text and have labelled every column.
> 3. The ordering of figure 1 and figure 2 has been fixed - now, figure 2 appears after figure 1.
> 4. We have updated Figure 3’s size of the text.
>
> ---
>
> Finally, we wanted to address the following question raised:
>
> > "In Eq. 2, is $s_0$ assumed to be fixed?"
> - We do not assume $s_0$, the initial state, is fixed. We only assume that we can sample multiple trajectories from this state. For example, we can sample $s_0$ from the initial state distribution and then sample multiple trajectories from it. We wanted to emphasize how we update the policy at this state in set RL versus in standard RL.
>
> ---
>
> Please let us know if we have misunderstood any question or any weakness you have pointed out - we are happy to clarify further and answer any other question you may have.

---

> > ### Comment · Reviewer_tLb8 · 2025-11-23
> >
> > I thank the authors for their careful response to my comments. I appreciate the explanation of entropy collapse, as well as the various edits to the figures. I also acknowledge that 3 seeds, while low, is sufficient in view of the computational requirements. As such, I have no further concerns, and I am happy to raise my score to a full accept (8).

---

### Official Review · Reviewer_5izw · 2025-10-31

**Soundness:** 2
**Presentation:** 3
**Contribution:** 4
**Rating:** 8
**Confidence:** 3

**Summary:**

This paper addresses the problem in RLFT (Reinforcement Learning Fine-Tuning) where policies lose diversity and "collapse" onto a handful of behaviors. It proposes a new problem, set RL, and improves policy gradient algorithms like PPO. For instance, in the advantage function calculation, it redefines a shared advantage term based on a "polychromic objective" (which simultaneously evaluates reward and diversity); it uses a "vine sampling" strategy to collect the on-policy data needed to evaluate the performance of trajectory sets. Experiments demonstrate that this method can effectively improve the policy's success rate, generalization ability to perturbations, and pass@n coverage.

**Strengths:**

The paper defines the "Set Reinforcement Learning" (set RL) framework , which generalizes the optimization of a single trajectory in traditional RL to the optimization of a set of trajectories . This framework is not only theoretically novel (for example, the authors derive a performance difference lemma applicable to set RL ), but it also provides a clear mathematical foundation that allows algorithms to directly optimize more complex objectives beyond standard rewards (such as the "polychromic objective" in this paper, which combines reward and diversity ). This holds significant inspiration and meaning for future work.

**Weaknesses:**

1. The definition of trajectory diversity is not general but must be engineered for each specific environment. For instance, it is defined as visiting different "sets of rooms" in BabyAI/Minigrid but different "sets of nodes" in Algorithmic Creativity.

2. To avoid exponential sampling complexity in long-horizon tasks, the method instead relies on 'vine sampling' . However, this sampling strategy itself imposes a major constraint, as it requires the ability to reset the environment to arbitrary states. This is unrealistic in most real-world robotics tasks, thus severely limiting its practical applicability.

3. The algorithm is "computationally demanding". This high workload stems from the multiple layers of sampling required, including the complex "vine sampling" procedure to gather sets of trajectories and the Monte Carlo sampling used to estimate the value baseline for the polychromic advantage. This makes the algorithm's workload substantially larger than that of standard PPO.

**Questions:**

1. I am concerned about the requirement for excessive sampling, which appears to be a significant overhead. Could the authors provide a computational complexity analysis for the algorithm?


2. Standard practice for evaluating RL algorithms often involves 'deterministic evaluation', where the action with the highest probability (i.e., argmax) is selected. If the environment is static, collapsing to a single deterministic solution is actually acceptable and often desired. In this context, wouldn't forcing diversity inherently degrade the model's optimal performance? Did the paper include any experiments using this standard deterministic evaluation to compare with the stochastic results?

---

> ### Author Response · Authors · 2025-11-22
> **Official Comment by Authors 1**
>
> Thank you for your thoughtful feedback; we have found the discussion on our weaknesses and the questions raised helpful.  We want to discuss these below.
>
> ---
>
> > “I am concerned about the requirement for excessive sampling, which appears to be a significant overhead”
>
> Our response to this question is similar to the response to Reviewer GrRC:
> - Note that all methods evaluated used the same number of rollouts per data collection step. All baselines used vanilla sampling (i.e., sampling $B$ full trajectories independently) while Poly-PPO used vine sampling which also samples $B$ trajectories in total. We explain this in Appendix A.1.1.  Our vine sampling scheme first generates 8 full rollouts, then for each rollout we select 2 states as our rollout states and then generate rollouts there. In total, we have 8 + 8^2(3-1) = 136 rollouts under vine sampling. We generate the same number of rollouts, $B = 136$, for REINFORCE, PPO and their UCB variants. This ensures that the comparison with other methods is fair in terms of data requirement.
>
> However, you raise an important question which is how crucial is it for poly-PPO to sample such a large number of trajectories?
> - Note that in principle, **we can run poly-PPO without vine sampling**. Assuming that the initial state $s_0$ is fixed, even with the standard/vanilla rollout scheme (where we just independently generate $B$ full trajectories), since all the rollouts start from the same $s_0$, we can apply the polychromic advantage to $s_0$. Intuitively, this ensures that the policy is exploratory near the initial state. Vine sampling is used simply to increase the number of states where we can apply the polychromic advantage and to potentially reduce critic bias [1].
> - We want to show, empirically, that the algorithm works well even without vine sampling i.e. when we sample a small number (8) trajectories. To do so, we test poly-PPO on the two most difficult tasks in BabyAI - synthseq and bosslevel. In both of these environments, we will generate only 8 independent rollouts (no vine sampling) for all methods and compare their performance - in this setting, the polychromic advantage can only be applied to the initial state (and across the polychromic window, see Appendix 1.1.1).  We report (avg reward, avg success rate) over 100 rollouts, 50 configurations, and 3 seeds.
> | Task       | REINFORCE                    | PPO                          | Poly-PPO                     |
> |------------|-------------------------------|-------------------------------|-------------------------------|
> | **Bosslevel** | (0.204, 24.9) ± (0.06, 6.72) | (0.202, 26.6) ± (0.05, 7.11) | **(0.267, 33.7) ± (0.06, 6.43)** |
> | **Synthseq**  | (0.236, 31.2) ± (0.014, 1.83) | (0.225, 26.8) ± (0.043, 3.960) | **(0.308, 38.9) ± (0.02, 1.15)** |
>
> ---
>
> > Standard practice for evaluating RL algorithms often involves 'deterministic evaluation... In this context, wouldn't forcing diversity inherently degrade the model's optimal performance?
>
> - This was a very helpful pointer and we would like to thank the reviewer for pointing us to carry out this experiment. We want to answer this question in two ways:
> - Firstly, our theoretical analysis shows that set RL, with *any* objective will naturally provide an implicit credit assignment to each trajectory individually (Proposition 5.1). We then demonstrated that heterogeneous sets that have more successful trajectories will attract more probability mass of the policy (Proposition 5.3). In other words, successful and unique trajectories are still stronger attractors of probability mass.
> - Experimentally, we see that even with deterministic sampling, poly-PPO matches/outperforms other methods. We look at pass@1 performance (similar to the paper’s experimental setup) but with deterministic sampling on BabyAI’s Bosslevel and Synthseq.
> | Task       | Pretrained Policy     | REINFORCE          | PPO               | Poly-PPO           |
> |------------|------------------------|---------------------|-------------------|--------------------|
> | BossLevel  | 22.4 ± 8.30            | 31.8 ± 2.40         | 36.0 ± 1.90        | **45.2 ± 2.30**        |
> | Synthseq   | 24.0 ± 2.45            | **36.0 ± 0.85**         | 32.0 ± 2.40        | **36.0 ± 0.23**        |
>
> ---
>
> References
> 1. VinePPO. arXiv:2410.01679

---

> ### Author Response · Authors · 2025-11-22
> **Official Comment by Authors 2**
>
> We would now like to discuss the weaknesses W1 and W2 (note that we have discussed the computational complexity of vine sampling in our previous comment and refer to the VinePPO paper [1] for further discussion).
>
> ---
>
> > The definition of trajectory diversity is not general but must be engineered for each specific environment.
>
> - This is completely valid and perhaps the most important way in which our method requires more work on the part of the practitioner than the baselines we compared with. In some settings, this can be made somewhat general. For e.g., in RL fine-tuning of LLMs on verifiable tasks, the diversity function could be an LLM-as-a-judge call i.e. asking an LLM judge to tell us how many of the $n$ generations are “semantically distinct”. However, this will almost certainly require some rubric, in-context examples, etc. In other cases, one might work with some empirical estimate of entropy as a proxy for diversity.
> - In our experiments, we relied on a hand-crafted diversity function to evaluate the performance of the policy itself while resting assured that the diversity function itself is accurate, which is certainly not an assumption we can make in real-world deployment settings.
>
> ---
>
> > To avoid exponential sampling complexity in long-horizon tasks, the method instead relies on 'vine sampling' . However, this sampling strategy itself imposes a major constraint, as it requires the ability to reset the environment to arbitrary states.
>
> - As pointed out in our previous comment, vine sampling is not a necessary component of our algorithm. However, if we do not incorporate vine sampling, then we *require* a fixed initial state in order to ensure that the polychromic advantage can be used at least at the initial state.
> - The requirement of a fixed initial state is, still, not trivial and is, indeed, a challenge to using our algorithm for robotic tasks.

---

> > ### Comment · Reviewer_5izw · 2025-11-25
> >
> > I would like to thank the authors for their detailed response. It has addressed most of my concerns. I have decided to maintain my original rating (8).

---

### Official Review · Reviewer_GrRC · 2025-11-01

**Soundness:** 3
**Presentation:** 4
**Contribution:** 2
**Rating:** 6
**Confidence:** 3

**Summary:**

This paper investigates a key limitation of reinforcement learning fine-tuning (RLFT) for pretrained policies namely, the collapse of behavioral diversity during fine-tuning, which leads to reduced exploration and exploitable outputs. The authors propose a new polychromic objective for policy gradient methods that explicitly encourages both exploration and refinement of diverse generations. They adapt proximal policy optimization (PPO) to this setting by introducing vine sampling for on-policy data collection and a modified advantage function consistent with the new objective. Experiments on BabyAI, Minigrid, and Algorithmic Creativity benchmarks show that the method improves success rates, generalization under perturbations, and coverage in pass@n evaluations. Overall, the work provides a principled approach to preserving and leveraging diversity in RL fine-tuning, addressing a major failure mode of current RLFT pipelines.

**Strengths:**

* I appreciate the time the authors spent to differentiate standard RL from set RL. The authors do a good job of establishing the setting before discussing the proposed polychromic algorithm.
* The results suggest that the polychromic algorithm is that set scoring is indeed useful at maintaining diversity while keeping the main performance/validity equal to that of PPO.
* This work carefully considers some of the flaws consistent in standard RL, entropy collapse and thoroughly discusses the possibilities of Polychromic PPO failure.
* I do think the contribution is a interesting novel extension of PPO.

**Weaknesses:**

* Perhaps the largest issue that the authors did not discuss is the time complexity of the generation of vines for the use of the algorithm, which might be the main trade off readers would consider when determining if the added diversity is worth the complexity cost. It may be the case that the methods were not explicitly fairly compared with respect to this trade off.
* Going a bit further on the initial point, would it be a fairer comparison to compare against other full Monte Carlo based algorithms that perform roll-out such as MCTS? Maybe I am mistaken, but is it reasonable not to include any runtime analysis against the proposed baselines?

**Questions:**

Is there anything that can be added besides the conclusion that can better inform the reader of the potential limitations above, or potential truncated sequences, as done by traditional monte carlo methods similar to n step bootstrapping?

---

> ### Author Response · Authors · 2025-11-19
> **Official Comment by Authors (1/2)**
>
> Thank you for your thoughtful feedback and questions. We respond to your questions below:
>
> ---
> > “The largest issue that the authors did not discuss is the time complexity of the generation of vines for the use of the algorithm” and “it may be the case that the methods were not explicitly fairly compared with respect to this trade off”
> - We would like to clarify that **we collected the same number of trajectories (during each on-policy data collection stage) for all methods**. Let us say that all methods are allowed to collect $B$ trajectories. The only difference between the rollout scheme, vanilla sampling, of the baselines and vine sampling [1,2] in our method is as follows. For the baselines with vanilla sampling, all $B$ trajectories were generated from the initial state of the environment. For vine sampling, some subset of the $B$ trajectories (i.e. the vines) are collected not by rolling out from the initial state but, rather, by resetting to some intermediate state visited in the rollouts already collected. We do this by ensuring that all methods collect the same number of rollouts, as explained in Appendix A.1.1. To illustrate this, suppose $B = 136$. We first generate 8 full trajectories, then we select 2 states from each of these trajectories as our rollout states and then generate trajectories from each of these rollout states. In total, we have $8 + 2\cdot 8^2= 136$ rollouts under vine sampling. We generate the same number of full trajectories for REINFORCE, PPO and their UCB variants. This ensures that the comparison with other methods is fair.
> - This also means that our method does not require more data than baselines since we are collecting the same number of full trajectories under all methods.
> - Furthermore, we would like to emphasize that the time complexity of vine sampling is not greater than baseline methods. If anything, resetting the environment to an intermediate state $s_t$ where $t \neq 0$ and generating trajectories from there onwards is more efficient than generating all trajectories from time $t = 0$. Of course, this is only possible in environments with resets, which is the setting we consider in our paper. We validate this claim with a wall-clock runtime analysis on an NVIDIA L40s GPU. Specifically, we compare the time taken by vanilla dataset generation and vine sampling to sample 136 rollouts. In other words, we sample 136 full rollouts (which we are calling vanilla sampling here) and 136 rollouts with vine sampling (every partial vine counts as a rollouts). Here, vanilla sampling takes $6.36 \pm 0.12$ s while vine sampling takes $5.24 \pm 0.88$ s to generate this dataset of 136 rollouts.
>
> There is, however, a secondary, orthogonal question which is important: does poly-PPO *require* sampling vine sampling or sampling a large number of trajectories?
>
> We want to answer this question both conceptually and through empirical validations.
> - In principle, **we can run poly-PPO without vine sampling** i.e. our algorithm can run with standard rollout sampling too. Assuming that the initial state $s_0$ is fixed, even with the standard/vanilla rollout scheme (where we just independently generate $B$ full trajectories), since all the rollouts start from the same $s_0$, we can apply the polychromic advantage to $s_0$. Intuitively, this ensures that the policy is exploratory near the initial state.
> - We want to show, empirically, that the algorithm works well even without vine sampling. To do so, we test poly-PPO on the two most difficult tasks in BabyAI - synthseq and bosslevel. In both of these environments, we will generate only 8 independent rollouts (no vine sampling) for all methods and compare their performance. We report (avg reward, avg success rate) over 100 rollouts, 50 configurations, and 3 seeds.
> | Task       | REINFORCE                    | PPO                          | Poly-PPO                     |
> |------------|-------------------------------|-------------------------------|-------------------------------|
> | **Bosslevel** | (0.204, 24.9) ± (0.062, 6.72) | (0.202, 26.6) ± (0.054, 7.11) | **(0.267, 33.7) ± (0.062, 6.43)** |
> | **Synthseq**  | (0.236, 31.2) ± (0.014, 1.83) | (0.225, 26.8) ± (0.043, 3.96) | **(0.308, 38.9) ± (0.022, 1.15)** |
>
> Then, what is the point of vine sampling?
> - Vine sampling allows us to use polychromic advantage at several timesteps which encourages exploration at various stages. For long-horizon tasks requiring several, sequential, high-level decisions across a rollout, this might be useful.
> - Vine sampling has been proposed even for standard PPO to help reduce the bias in the learned critic [2]. We expect vine sampling to be even more useful in reducing critic bias for long-horizon tasks. Insofar as we expect this sampling scheme to be more widely used, we wanted to show that our algorithm can further benefit from vine sampling.
> ---
> References:
> [1] Trust Region Policy Optimization. arXiv:1502.05477
> [2] VinePPO. arXiv:2410.01679

---

> ### Author Response · Authors · 2025-11-19
> **Official Comment by Authors (2/2)**
>
> In addition to our earlier comment, we would also like to address some other questions raised.
>
> ---
> > would it be a fairer comparison to compare against other full Monte Carlo based algorithms that perform roll-out such as MCTS?
>
> Since MCTS is an inference-time (or test-time) search algorithm, as opposed to as an on-policy RL training algorithm, we believe this is not a fair comparison. Our paper is on algorithms for reinforcement learning that enables both higher success and diversity/exploration. MCTS and other inference-time algorithms that use compute at test-time for decision-making can be used on top of our algorithm as well. In particular, one of the strongest motivations for training an agent to solve a problem in diverse ways, as mentioned in our introduction, is so that the agent benefits even further from having access to test-time compute - if there is no diversity in the generations of a model at test-time, the availability of MCTS or other such algorithms does not make much of a difference. Indeed, the pass@n curves, which can be interpreted as using an oracle verifier for test-time compute, supports this claim.
>
> > Is there anything that can be added besides the conclusion that can better inform the reader of the potential limitations above
> - There are several potential limitations and we view this submission as a first attempt in what we hope will be a long line of work investigating set RL and objectives that train some version of exploration-exploitation trade-off. We discuss two more limitations (on top of the conclusion section of our paper) here.
> - Note that we use the same polychromic objective in our set RL framework throughout training. This means throughout the entire training period, our policy is trained to optimize for both high diversity and high rewards. However, if one uses a diversity function that is directly opposed to reward, this could be problematic. As an example, suppose for a math problem, we have two solutions $y_1$ and $y_2$. Suppose, $y_2$ essentially has the same high-level strategy as $y_1$ but makes an arithmetic error in between that leads to a different final solution. A diversity function that calls these two generations distinct may incentivize diversity in the wrong manner, i.e., diversity in answers to a simple arithmetic question which has a universally correct answer and should not result in diverse solutions. This illustrates that while our work assumes access to a diversity function used in several prior works [3,4,5,6], the construction of such diversity metrics is not trivial and poly-PPO is certainly not completely immune to inaccurate/ineffective diversity functions, much like how we do not expect RL algorithms to be immune to poorly designed reward functions/verifiers.
> - Similarly, while we discussed the importance of discovering diverse methods of solving a task, this certainly is not necessarily true for all tasks (such as solving simple arithmetics). As such, there is an important question one must consider before choosing between poly-PPO and other RL algorithms (such as PPO, REINFORCE, etc.) regarding whether we want the agent to explore in the task we are concerned with in the first place.
>
> ---
>
> Please let us know if we have misunderstood any question or any weakness you have pointed out - we are happy to clarify further and answer any other question you may have.
>
> ---
> References:
> [3] NoveltyBench. arXiv:2504.05228
> [4] Jointly Reinforcing Diversity and Quality. arXiv:2509.02534v1
> [5] Vendi Score. arXiv:2210.02410
> [6] Semantically diverse language generation. arXiv:2406.04306

---

> ### Author Response · Authors · 2025-11-27
> **Follow Up**
>
> Dear Reviewer GrRC,
>
> We wanted to follow up since there is less than a week left in the rebuttal period. We would like to know if our rebuttal above has addressed your questions/concerns. We would be happy to answer any remaining questions and discuss further or provide more evidence either through empirical validations or theoretical analysis.
>
> So far, in our rebuttal above, we have done the following:
> - **addressed the concern regarding time-complexitiy of vine sampling**: we clarified that in our vine sampling set up, we did not sample more trajectories than the standard rollout sampling method, and the time complexity is not larger.
> - **clarified that all RL methods used the same number of rollouts, with or without vine sampling**: this was to clarify that all methods were, indeed, fairly compared in our empirical results.
> - **demonstrated that polychromic PPO can work *without* vine sampling as well**: we clarified the role of vine sampling, showed that conceptually poly-PPO can work without vine sampling as well if the initial state is fixed and demonstrated empirically that poly-PPO still achieves strong performance without vine sampling.
> - **discussed additional limitations of our method**.
>
> If all your questions and concerns have been addressed, we would be grateful if you could acknowledge that.
>
> Thank you,
>
> Authors

---

### Comment · Area_Chair_vm7D · 2025-11-21
**Author-Reviewer Discussion**

Dear reviewers,

Please review the authors' response and adjust your rating accordingly. If you have any further questions, please discuss with the authors further.

AC

---

### Author Response · Authors · 2025-11-29
**General Response and Manuscript Updates**

Thank you to the reviewers and ACs for engaging in the review process for our paper. We summarize the key points of discussion and changes we have made to the manuscript here.

---

``Question: How expensive is vine sampling?``
- **All methods were tested and compared under the same data budget** i.e. every method collected the same number of trajectories during each data collection step – we clarified this in our response to Reviewer 5izw and Reviewer GrRC.
- **Vine sampling does not increase the wall-clock time required for the data collection step** - we clarified this in our response to Reviewer GrRC both conceptually and empirically.
- **Vine sampling collects strictly *fewer* state-action tuples*. This is because, under vine sampling, not all trajectories start from the initial state - the vines start from states at intermediate time-steps.

---

``Question: Is vine sampling necessary for polychromic PPO?``
- **Vine sampling is *not necessary* for polychromic PPO. If the initial state can be fixed, poly-PPO is compatible with standard rollout scheme as well** – explained this in the response to Reviewer GrRC and 5izw both conceptually and empirically. Our method requires us to be able to sample multiple trajectories from some state so we can apply the polychromic advantage; if the initial state is fixed, then we can naturally apply this to the initial state.
- **Empirically, even without vine sampling and under a small trajectory budget, polychromic PPO still achieves strong performance and outperforms the baselines**. We ran poly-PPO, REINFORCE and PPO on BabyAI’s *Synthseq* and *Bosslevel* and only collected 8 trajectories at each data collection for all methods. We saw that poly-PPO’s performance degradation is less steep.
- **Vine sampling is incorporated only to encourage exploration at multiple stages of a rollout, which can be useful for long-horizon tasks**. Vine sampling simply ensures that there are several states, in addition to the initial state, at which we can apply the polychromic advantage to encourage exploration. For long-horizon tasks that require exploration at multiple stages, this can be quite helpful. Also vine-sampling also enables standard RL algorithms to perform better by reducing critic bias [1].

---

``Question: Does the polychromic objective compromise on pass@1 performance or deterministic evaluation?``
- As is evident from the pass@$k$ curves (Fig 1 and Fig 3) and from our deterministic evaluation (as demonstrated to Reviewer 5izw, poly-PPO does not significantly compromise on either when it has discovered some solution during the RL training.
- In the bandit setting, this is in line with our theoretical analysis that shows that successful trajectories get an implicit credit assignment. In Proposition 5.3, we see that sets with more successful trajectories are stronger attractors of the policy’s probability mass. In Proposition 5.2, we see that even homogeneous sets of successful trajectories are attractors of probability mass despite their set value being 0 under polychromic objective.
- In the non-bandit setting, the additional reason for this being true is that the polychromic advantage is not applied to all states - it is only applied to rollout states. At every other state, the objective is the standard PPO which is an exploitation objective. The interpretation here is that, at rollout states, our algorithm encourages exploration and then, at subsequent time-steps, the algorithm encourages exploitation.

---

``Question: How is set RL different from multi-objective RL (MORL)?``
- **Set RL can be used to optimize a single objective**. For example, one can set $f(s, \tau_1,\cdots, \tau_n) = \frac{1}{n}\sum_{i=1}^n R(\tau_i)$ to optimize a “set-REINFORCE” objective or optimize $f(s, \tau_1,\cdots, \tau_n) =\max (R(\tau_1),\cdots, R(\tau_n))$ optimize a pass@$n$ objective.
- **Set RL optimizes objectives over a set of objectives and the credit assignment is *shared* across all the trajectories, as opposed to individualized credit assignment for each trajectory**. This is due to both the objective being defined over multiple trajectories and the baseline not discriminating between trajectories within the set. Both these differences (and other observations) were pointed out to Reviewer Bmic.

---

We have made several changes to our manuscript (highlighted in blue), including additional discussion on the motivation for entropy analysis, differences between set RL and MORL, changes to the figures for improved readability, etc.

---

References:

[1] VinePPO. arXiv:2410.01679

---

### Meta-Review · Area_Chair_seAn · 2025-12-22

**Summary:**

Reviewers raised concerns regarding the complexity, necessity, and underlying mechanism of the vine sampling procedure introduced in this paper. These concerns were well addressed in the authors’ rebuttal. Notably, all method variants use the same data budget, and vine sampling does not increase wall-clock time, alleviating concerns about computational overhead.

The only remaining concern relates to the absence of a multi-objective reinforcement learning (MORL) baseline, since Set RL optimizes both reward and diversity simultaneously. While including such a baseline would further strengthen the paper, this omission does not substantially diminish the overall contribution. The authors have clarified the conceptual differences between Set RL and MORL in the rebuttal. Overall, I recommend acceptance.

**Reviewer Concerns:**

### Addressed
* **Complexity and necessity of vine sampling**
The authors provided detailed explanations of the motivation, mechanism, and benefits of vine sampling, which adequately address reviewers' concerns regarding its complexity and necessity.

* **Difference between Set RL and MORL**
The authors included a detailed discussion clarifying the conceptual and optimization differences between Set RL and MORL, resolving confusion raised by the reviewer.

### Outstanding
* **MORL as a baseline**. Reviewer JQLZ raised the concern that a MORL baseline would be an appropriate comparison. This remains a missing experimental component, and including such a baseline would further strengthen the paper. However, given the constraints of the current review process and the lack of a specific suggested MORL method during discussion, this issue is not considered critical to the paper's main contribution.

**Reviewer Scores:**

* **Reviewer GrRC, Reviewer 5izw, Reviewer tLb8 (initial score: 6, 8, 6)**. These reviewers were generally positive, with concerns primarily focused on clarification and minor experimental details. The rebuttal addresses these points effectively, so I would expect these reviewers to maintain or slightly increase their scores after discussion.

* **Reviewer JQLZ (initial scores: 2)**. This reviewer raised a major concern regarding the relationship between Set RL and multi-objective reinforcement learning (MORL), and suggested incorporating MORL as a baseline. Based on the rebuttal and discussion, the reviewer should now have a clearer understanding of the distinction between Set RL and MORL. The remaining concern is the lack of a MORL baseline. I would therefore expect the reviewer to either maintain their initial score or increase it modestly, depending on the importance they place on the missing comparison.

---

### Decision · Program_Chairs · 2026-01-26

Accept (Poster)